# Reliable Vehicle Routing Problem Using Traffic Sensors Augmented Information

**DOI:** 10.3390/s25072262

**Published:** 2025-04-03

**Authors:** Ahmed Almutairi, Mahmoud Owais

**Affiliations:** 1Department of Civil and Environmental Engineering, Majmaah University, Al-Majmaah 11952, Saudi Arabia; a.alaoni@mu.edu.sa; 2Civil Engineering Department, Faculty of Engineering, Assiut University, Assiut 71515, Egypt; 3Civil Engineering Department, Faculty of Engineering, Sphinx University, New Assiut 71684, Egypt

**Keywords:** stochastic routing, deep learning, traffic sensors, traffic flow estimation, vehicle routing optimization

## Abstract

The stochastic routing transportation network problem presents significant challenges due to uncertainty in travel times, real-time variability, and limited sensor data availability. Traditional adaptive routing strategies, which rely on real-time travel time updates, may lead to suboptimal decisions due to dynamic traffic fluctuations. This study introduces a novel routing framework that integrates traffic sensor data augmentation and deep learning techniques to improve the reliability of route selection and network observability. The proposed methodology consists of four components: stochastic traffic assignment, multi-objective route generation, optimal traffic sensor location selection, and deep learning-based traffic flow estimation. The framework employs a traffic sensor location problem formulation to determine the minimum required sensor deployment while ensuring an accurate network-wide traffic estimation. A Stacked Sparse Auto-Encoder (SAE) deep learning model is then used to infer unobserved link flows, enhancing the observability of stochastic traffic conditions. By addressing the gap between limited sensor availability and complete network observability, this study offers a scalable and cost-effective solution for real-time traffic management and vehicle routing optimization. The results confirm that the proposed data-driven approach significantly reduces the need for sensor deployment while maintaining high accuracy in traffic flow predictions.

## 1. Introduction

The problem of finding the optimal route/path between an origin-destination (OD) pair in transportation networks (i.e., RTNP) is always a crucial task for both passengers and operators. Typically, operators investigate this type of issue in order to efficiently manage the network. The operators are not concerned with transporting each passenger via the shortest route but rather with maintaining the network’s overall harmony. This is evident in the design of transit routes, as evidenced in [1,2,3]. The generated routes must also be the quickest way to transport users from their origin to their desired destination. Regarding the two perspectives, the methodologies for creating the route choice set for each O/D pair have been divided into riders’ perspectives and operator’s perspectives. This conflict is evident in the work in which generated routes deviate from the shortest path to alleviate network congestion while attempting to maintain passenger acceptance [4,5].

In addition, the RTNP changes when the generated routes reflect the perceptions of the network passengers. Due to a lack of network information/familiarity, the majority of users do not always choose the shortest path [6]. Therefore, the problem becomes a route selection problem, as in traffic assignment approaches; consequently, there are various methods for generating routes based on the chosen traffic assignment method [7,8,9]. In contrast, the real-time shortest path is the ultimate goal that must be generated upon request (e.g., Google Maps).

The RTNP data availability and the stochastic network representation also contribute to the problem complexity. The literature has several levels of network representation, ranging from naive to complicated [10]. The most important aspect of route-finding algorithms is the description of street/edge transit time. When the travel time on arcs is deemed predictable (invariant and certain), the Dijkstra algorithm [11] readily finds the shortest path. In reality, however, each link’s journey time is variable and difficult to anticipate. It varies with the level of traffic on it; therefore, the algorithm cannot discover the optimal path until all links’ traffic is monitored in real time, which is impractical [12,13,14]. The traffic is also stochastic with respect to time, resulting in travel time uncertainty and transforming the RTNP into what is known as a Stochastic Routing Transportation Network Problem (SRTNP).

Typically, the problem of travel time stochasticity is simplified by assuming the presence of a probability density function (PDF) for each network edge. Then, any of the deterministic shortest-route approaches [15] may utilize the anticipated journey time along each arc. More complexity can be induced since this variability/stochasticity can be seen as non-stationary random variables [16]. Travel time on streets is a result of traffic flow, which in turn is determined by origin-destination (OD) flows. The flow of links/streets can be estimated using traffic assignment models, provided that an accurate OD matrix is available or can be observed through traffic sensors. However, deploying traffic sensors on all streets is not a practical task. In cases where travel time estimation from traffic flows—either in real-time or for future predictions—is not a straightforward task, alternative methods must be considered.

This study addresses the SRTNP by generating a set of stochastic routes for each node pair (OD). Each route in the set may be the shortest path in this dynamic environment. A method based on repeated simulations is created to offer a set of non-dominated pathways since none of them can be deleted from the selection set. Then, a traffic sensor location (TSLP) algorithm is used to ensure that the threshold of the sensors covers each edge of the generated path’s. Finally, a machine learning (ML) algorithm is used to predict traffic volumes on unobserved transportation network edges. This methodology would enable any shortest path to provide the rider with the optimal route when their journey is about to begin. The maintained percentage of sensors on each path would provide considerable reliability to the estimated travel time, in addition to the accuracy of the ML predictions.

The remainder of this study is structured as follows: Section 2 presents a concise overview of the existing research and highlights the knowledge gap addressed in this study. Section 3 outlines the fundamental principles and theoretical framework that underpin the proposed methodology. Section 4 details the developed algorithms and solution methodology, including the integration of traffic sensor data, deep learning models, and stochastic routing strategies. Section 5 evaluates the effectiveness of the proposed approach using real-world transportation networks and demonstrates its applicability and performance. Finally, Section 6 presents the conclusions and future research directions.

## 2. Literature Review

This section provides an overview of the state-of-the-art of the three methodology pillars: routing problems, TSLP, and ML techniques. Then, it would be apparent that incorporating them into a single framework to solve the formulated SRTNP will be the salient contribution of this article.

### 2.1. Routing Problem

This subsection discusses the basic assumptions that are often made to solve the routing problem. Numerous applications in diverse sectors, including communications, transportation, and electrical networks, face the route-generating challenge [17]. In deterministic networks, the shortest path is the foundation of all routing methods. One of the complications added to the problem is that travel times on edges are dependent on the flow, with a function that connects the delay to flow. Therefore, the methods for designating the shortest path are changed to include this additional component of complexity [18]. This assumption is still inadequate for the SRTNP because it is impossible to locate these types of stable time-dependent functions in the real world. Considering the conditional probability between the time and surroundings of an edge; reference [19] provided a dynamic dimension to the problem. A Markovian representation aided in determining, the appropriate policy for advancing along the considered arc or waiting until conditions improve at each node. Even though the network horizon is a continuous interval, discretizing it into time slots is considered the most effective technique for reducing the complexity [20]. The slots were connected via a time-dependent function that calculated the travel time for each edge based on the arrival time at the start node. It is an extension of the traditional deterministic shortest path problem, which can be solved using Bellman’s equation [21]. Unfortunately, Bellman’s recursive principle of optimality is invalid in the case of random trip periods.

Therefore, the deterministic time-dependent assumption becomes inapplicable in transportation networks, where there is an increase in trip time uncertainty (stochasticity). This unpredictability is caused by fluctuating demand, lack of information, and traffic accidents [22,23]. It not only minimizes the trip durations of the created routes but also increases the dependability of the routes’ in the face of such uncertainties. The groundbreaking reference [24] focused on locating the shortest route in networks with time-dependent random trip durations. The concept of determining the route with the least anticipated travel time (LAT) is resolved using an iterative branch-and-bound approach. When addressing adaptive path selection, the described problem might be applied to the hyper-path dilemma, as observed in transit assignment problems. In [16], label-correction techniques were applied to search for LAT routes. However, preventing routes with cycles cannot be ensured without retaining the first–in–first–out (FIFO) condition, which is not a reasonable assumption for the SRTNP procedure. FIFO dictates that for every edge (s–d), no one can pass vertex (d) before someone has already passed node (s) [25]. This characteristic negates the reality of vehicle overtaking. It also precludes the possibility that departing destination later may enhance the total trip time [10,19].

There is a widespread perception that adaptive routing can continually improve the travel time of a journey. Using journey time expectations as random variables, the SRTNP determines the LAT routes before executing the trip. Then, some information on the following link’s transit time may be disclosed during the voyage (en route). Then, a process of rerouting might be conducted based on this knowledge [26,27]. Adaptive routing is employed when the actual arrival time at the link’s start node is discovered en route [28]. If the travel times on links are expected to be dependent on the arrival time at the link’s starting node.

Several researchers have disregarded the association between edge travel times by addressing the problem in its generic form without identifying it as an SRTNP [29]. Some research assumed the presence of a relationship owing to traffic accidents or severe weather in order to analyze this characteristic. Consequently, they restricted it to the next link status in a limited manner [30,31]. Even research that examined the extended correlation between all STN linkages required the existence of a discrete link time joint distribution or a covariance matrix. They did not, however, offer a rational mechanism to obtain this knowledge in the real world [32,33,34,35]. Using Monte Carlo (MC) simulation to draw distinct samples for different trip time realizations was the optimal method for driving the path time distribution from the preset edges’ correlation. Under the premise of an independent link-passing time, simulation is not required [36].

Alternatively, research [37,38,39] focused on the multi-objective aspect of the problem, in which the estimated travel duration is not the only factor for selection. Other parameters, such as the path’s fluctuation period and dependability, can be evaluated.

The association between travel time and demand on a network’s origin-destination (OD) pairings is primarily the result of a complex interrelationship between the available route option set and the flow-dependent travel times of links. Fortunately, traffic assignment models can easily manage this correlation and provide a temporal covariance matrix for the edges [40,41].

The computational cost is the greatest obstacle for the majority of routing algorithms, given that users request routes in real-time. Even under stringent assumptions, existing routing formulations have been proven to be NP-hard. Therefore, offline planning may be a viable option for evaluating this issue. Due to advancements in server technology, it is now possible to utilize previously established routes and store them for online use. In the planning phase, consecutive demand simulations may reflect anticipated network circumstances [5].

In [10], the offline SRTNP is tackled in a multi-objective framework. In contrast to previous approaches that assumed a known PDF to monitor variability in journey time, the correlation between traffic flow and the time of related links was used to extract trip time variability. The time horizon is partitioned into time intervals or slots, and it is postulated that the network operates under a static traffic equilibrium but with varying traffic circumstances for every time slot. The suggested technique begins with demand information obtained in advance, previously produced pathways, and a selected traffic assignment algorithm. Numerous simulations were performed at each network interval. It generates PDFs for the trip times of both routes and streets, taking into account the correlation between them. The multi-objective analysis revealed that the lowest mean travel time path is not necessarily the most reliable path to be selected/generated.

### 2.2. Sensor Location Problem

Efficient traffic management, control systems, and transportation planning rely on accurate data regarding traffic flow. Typically, these data are gathered using traffic sensors; however, deploying sensors across every link in a network is neither practical nor feasible. Instead, it is necessary to extrapolate information from a limited set of observed link flows to estimate the overall network traffic. Most research has focused on determining the optimal number and placement of traffic sensors, either for monitoring link flows or estimating origin-destination (O/D) demand. Traffic observation aims to precisely infer flow across all links by analyzing network structures and user behavior, such as turning ratios at intersections and link-path incidence matrices. In contrast, traffic estimation focuses on reconstructing the O/D matrix using a reference dataset and statistical assumptions, such as conditional probability distributions to account for unknown variables. Studies [42,43] provide a more refined classification of TSLP. Across all TSLP approaches, research primarily seeks to optimize the sensor placement and quantity to either reduce costs or enhance estimation accuracy [44,45,46].

The sensor type was found to be a dominant factor in the problem solution search. Two main types of sensors for counting purposes can be identified: passive and active sensors. In passive counting mode, only the link volumes are recorded, whereas the active mode enables the collection of additional data using specialized sensors. These include path-ID sensors for route tracking, vehicle identification sensors for monitoring travel times, and image sensors for classifying vehicle types [47,48,49]. Sensor data can either be assumed to be error-free or contain measurement inaccuracies [50]. This study focuses on passive-mode link sensing, which presents a greater challenge compared to other methods since it provides minimal traffic information—only the count of vehicles passing through a link. However, it remains the most widely used and cost-effective traffic-monitoring approach [51,52].

In general, the TSLP for any network is defined by the constraint that only a limited number of links can be equipped with counting sensors. The challenge lies in extending the available link flow data with the highest possible accuracy. The relationship between node pair demand and edge flow follows the well-known assignment mapping equation, expressed as follows(1)X=∆PDwhere; X is the vector of observed link flows, D is the vector of node pair demands, P is the path-demand proportion matrix, and ∆ is a link-path incidence matrix. Equation (1) represents the capability of calculating the demand for node pairs from the traffic counts on links. Provided that the number of independently observed links is equal to the number of unknown demand pairs, or in other words, the resulting matrix [∆*P*] is of full rank. The stated equation allows solving the traffic estimation problem mathematically, i.e., calculating the vector *D* and then solving the assignment problem to obtain the complete link vector.

Unfortunately, the number of OD pairs is usually larger than the number of observations. Therefore, the problem is underspecified, and there is an infinite set of solutions for the OD demand pairs to satisfy Equation (1) [53]. This condition has led many studies to combine prior information with observations to obtain a unique solution. The prior information is a reference OD matrix, which usually comes from many different sources, including an old out-of-date OD matrix. The existence of such a matrix is a common assumption in the literature [41,53,54,55,56,57,58].

Moreover, the location and number of sensors are the most important criteria for controlling the estimated flow accuracy. Therefore, many studies have attempted to solve the location problem along with the flow estimation problem [41,48,56,58,59,60,61,62,63]. The well-known location rule accepted by all these studies is the covering rule presented firstly in [59,64]. It stipulates that the sensors should be installed on the edges to intercept at least one portion of each node pair flow.

Through four binary integer programming formulations, sensors were dispatched in [65] to intercept the OD nodes in a network. A genetic algorithm (GA) technique was then used to solve the provided formulas. However, the accuracy of the predicted OD matrix using the route estimator approach was not accounted for in the GA fitness function. In [66], the same integer programming concept is used for time-expanded networks. Studies [58,66] used a generalized least-squares formula with a Kalman filtering approach to create optimal OD pair dependability using maximal information benefits. To checking for the inaccuracy of the estimated matrix, Heuristics were provided for selection based on Bayesian network convergence in [41]. The authors of [67] expanded their work in [66] by transforming the issue into a multi-objective model that considers OD pairing coverage and edge information gains. Finally, they solved it using a Hybrid Greedy Randomized Adaptive Search Procedure. In [12], the TSLP was introduced as a min-max optimization model to minimize the maximum deviation of the goal OD matrix. While reference [68] enhanced the method of [12] by taking into account the influence of time-dependent sensor failure throughout the sensor’s lifespan in the position-setting technique.

Alternatively, traffic flow information can be extended by applying the conservation law at network nodes, which states that the total outflow must equal the total inflow. In [55], a methodology was introduced for strategically placing a subset of sensors at network nodes to derive the complete link flow vector, assuming that the turning ratios at nodes are predetermined and fixed. Similarly, in [69], the sensor placement was optimized by determining both the location and number of sensors required on network links. This approach also relies on the conservation law to infer all link flows, with the assumption that the link-path incidence matrix is known in advance.

In [70], the work of [69] was extended to infer all link flows without the requirement of path enumeration. Then, in [71], using the same inference technique, the minimum number of links was searched to be observed for inferring a desired subset of link flows rather than all the flows. In a similar way, in [72] the problem of locating active sensors (path-ID sensors) on the arcs is addressed to determine route flow volumes and, consequently, all link flows using a link-path incidence matrix. Edge flow inference can be considered a combined problem of the TSLP. It takes into account the lack of prior knowledge of route choice proportions.

In [73,74], the TSLP was investigated from a combinatorial perspective to identify the minimal number of sensors on the nodes, given the availability of turning ratios, to deduce the link flows using the conservation equations. The issue is shown to be NP-complete in a variety of situations, with the exception of a reduction to a polynomial via an unusual-graph representation. Alternatively, ref. [75] circumvented the computational cost of these algebraic approaches by using a graphical method based on a minimal spanning tree. A novel linkage solution technique for this issue has gained popularity [50,76]. Determining the upper limit of the solution to the problem is a straightforward process. In [77], the idea of new connections was used to develop two distinct goal functions for mitigating the impact of the sensor failure probability on the link inference range. Additionally, the study in [78] used the same idea while reducing the error propagation of the collected counting measures for every link inference. Interestingly, reference [14] produced remarkably similar findings to these identical link flow inference approaches with deep learning neural networks and fewer sensor counts. The suggested method can learn the latent associations between the flow components of a network to accurately forecast missing link data. They did not, however, show a solid geographical plan to expand their contributions.

Recently, reference [79] devised a mathematical analytical approach combining backward substitution and a factorization scheme to determine the number of observations required to solve either the whole or partial observability flow estimation problem. However, network scalability is a challenge for the proposed strategy. In [80], the formulation was relaxed into binary linear programming by enumerating all possible paths. The issue was then solved using a relaxed variant of linear programming with a manageable time complexity. Alternatively, in [13], the same formulation was heuristically handled by spreading traffic emission remote sensing monitors to collect prohibited emissions in a network. In [81], three-screen line formulas for portable excess speed sensors were developed. For investigations [12,13,43,81], the suggested Heuristics intercept a certain number of pathways (e.g., three to seven routes per OD), but not all routes in the examined networks. This route generation stipulation matches the fact that the majority of traffic in a network is carried by at most five routes per OD [68,82,83,84].

### 2.3. Machine Learning

Machine learning (ML) is an intelligent tool for predicting or estimating unknown variables based on learning paradigms. ML approaches have been applied to email filtering, computer vision, large language models, and speech recognition since it is too expensive to design algorithms to accomplish the required jobs. In general, the accuracy of ML models is much greater than that of traditional statistical approaches. However, the offered ML models are not interpretable since, in the majority of situations, they are viewed as “black box” tools [85].

Artificial neural networks (ANNs) are among the most well-known ML approaches. Their structures have evolved over time to provide more sophisticated learning procedures, and they are now extensively used. Architectures based on deep learning (DL) are now regarded as the most frequent models in transportation science applications. If we consider the vehicle routing problem using traffic sensor information, the authors are unaware of any ML application for the topic [86,87,88,89]. ML methods are limited to forecasting the flow of network edges as time-series problems. Dynamic temporal relationships contribute to the challenges of traffic forecasting. Recently, some research has also addressed the spatial correlation of the network among intersection/street flows in order to correlate observable and unobserved sites [14,90]. Although these studies are devoted to static transportation network cases, they can easily be extended to the dynamic case in which the edge flow is time-dependent.

Furthermore, data fusion techniques have been explored to enhance the route flow estimation. Several recent studies have demonstrated the effectiveness of deep learning-based models for traffic state estimation. Reference [91] introduced an end-to-end heterogeneous graph neural network (GNN) model for traffic assignment that efficiently captures network-wide flow distributions even under incomplete data conditions. Their results highlighted the superior adaptability of graph-based architectures in modeling complex traffic interactions compared with conventional methods. In a related study, reference [92] proposed a deep learning-based dynamic traffic assignment framework that integrates incomplete OD data into a predictive model, offering a scalable alternative to traditional equilibrium-based assignment techniques. This study demonstrates that deep neural networks can effectively mitigate data sparsity issues and improve dynamic routing performance under stochastic conditions. Beyond OD-based flow estimation, reference [93] explored a data fusion approach by combining probe vehicle trajectory data with automated vehicle identification (AVI) records to reconstruct route-level traffic flows. Their findings emphasized that the integration of multi-source traffic data significantly enhances the accuracy of flow prediction models, particularly for urban networks with sparse sensor coverage. Additionally, reference [94] extended this approach by developing a 3D convolutional neural network (CNN) model that estimates dynamic OD flows using AVI data, leveraging spatial-temporal correlations to improve traffic state reconstruction in real-world networks. Their results underscore the benefits of deep learning architectures for capturing complex temporal dependencies and enhancing OD flow estimation accuracy.

Recent studies have demonstrated that graph neural networks (GNNs) are highly effective for modeling dynamic traffic states by capturing the spatial dependencies between road segments [95,96,97]. Reference [98] provides a comprehensive survey on graph neural network methodologies for spatiotemporal data modeling, highlighting their ability to integrate sensor data for real-time traffic forecasting. Similarly, reference [99] explored urban region profiling using ST-GNNs, demonstrating the model’s ability to infer missing link flows based on connectivity patterns and traffic variations. In addition to GNNs, hybrid graph-based deep learning models have been increasingly used to improve traffic-flow estimation. Reference [100] introduced a Spatial-Temporal Graph Attention Gated Recurrent Transformer Network, which combines GNNs with attention-based architectures to enhance long-term traffic flow forecasting. Their results indicated that graph-based representations outperform traditional time-series models, especially when dealing with highly dynamic road networks. Alternatively, reference [101] presented a Hierarchical Spatio-Temporal Graph Convolutional Transformer Network, further demonstrating the robustness of hybrid GNN-Transformer models in capturing multi-scale spatial-temporal patterns in transportation systems. Beyond graph-based models, Transformer architectures have recently gained traction in traffic prediction and stochastic routing problems due to their ability to model long-range dependencies and sequential variations in traffic flow. Reference [102] introduced a Spatio-Temporal Parallel Transformer (STPT) model that effectively captures traffic dynamics across large-scale urban networks, outperforming recurrent and convolutional models in terms of predictive accuracy. Reference [103] proposed a Transformer-Based Spatiotemporal Graph Diffusion Convolution Network, demonstrating that self-attention mechanisms can significantly improve traffic state estimations, particularly in cases of sparse sensor availability. Reference [104] leveraged Spatio-Temporal Graph Transformers for fine-grained data analysis, showcasing their applicability in recognizing complex travel patterns. In large-scale transportation systems, references [105,106] proposed a Transformer-based Spatio-Temporal Traffic Prediction model for metro networks, while reference [107] introduced a Graph Spatial-Temporal Transformer Network that enhances real-time traffic prediction in intelligent transportation systems.

To this end, we emphasize the existing gaps in the literature and the work’s potential contribution. This study addresses a significant gap in the SRTNP literature by challenging the prevailing assumption that adaptive routing consistently leads to optimal travel times. Traditional approaches to vehicle routing in stochastic transportation networks rely heavily on real-time adjustments, often overlooking the inherent variability of link travel times and the potential for suboptimal routing decisions. By establishing a preplanned multi-objective routing framework, this study provides an alternative strategy that accounts for both the expected travel time and reliability considerations.

We introduce the concept of sensor data augmentation, which refers to the generation of synthetic traffic flow data to compensate for missing or sparse sensor data. Unlike traditional data fusion paradigms, such as feature concatenation or attention-based cross-modality interaction, the proposed approach leverages deep learning models to learn robust representations from incomplete traffic data. The goal is to fill in the missing data by capturing the spatial and temporal dependencies inherent in the traffic network rather than simply aggregating data from various sensor sources. Feature concatenation typically combines data from multiple sensors into a single feature vector, while attention-based interaction focuses on aligning multi-modal data through learned attention mechanisms. In contrast, we focus on enhancing sparse sensor data by learning the underlying traffic patterns directly from the data, thereby augmenting the missing information without relying on traditional data aggregation strategies. This approach allows us to preserve important traffic flow patterns while filling in gaps where data may be missing or incomplete.

The novelty of this work lies in the integration of deep learning techniques into traffic sensor-based information augmentation, an aspect that has been largely unexplored in previous research. While existing studies on the TSLP focus primarily on deterministic flow estimation, they do not incorporate data-driven learning models to infer missing link flows. This study introduces a hybrid approach that optimally deploys a minimal set of traffic sensors and enhances their coverage using DL-based flow estimation models, thereby bridging the gap between limited sensor availability and complete network observability.

Furthermore, this study advances the state-of-the-art in stochastic routing by demonstrating that route selection in uncertain environments should not be based solely on the shortest-path heuristics. Unlike conventional methods that treat travel time variability as a secondary factor, the proposed framework explicitly integrates travel time uncertainty into the route selection process. This ensures that users are provided with routing alternatives that balance speed and reliability, which is a critical aspect that has been largely overlooked in prior formulations.

## 3. Problem Formulation

### 3.1. Network Representation

In this work, the STN is represented by the graph *(N, A, Ŵ, Θ, ζ)*, where *N* is the group of nodes/vertices linked by directed edges/arcs/links/streets *A = {a_ij_: i, j*
∈N*}*. A predefined number of vertices called demand vertex pairings with origin (*s*) and destination (*r*) are collected in the vector *Ŵ = {1*, *2*, *…*, w, *….*, *ŵ}^t^*, where *t* stands for transposition. The analytical time horizon *Θ* is discretized into the set *Γ = {t*_0_, *t*_0_
*+ δ*, *t*_0_ + *2δ*, *…*, *τ*, *…*, *t*_0_
*+ nδ}* with the time period unit *δ*, during which the network experiences a transient static equilibrium caused by the demand vector *D_τ_ = {*$d_{\tau }^{w}$*:*∣ *D_τ_* ∣ = ∣*Ŵ* ∣ = *ŵ}*, which is correlated with the specified time slot (*τ*). *ζ* is the set of all arc travel times = *{*$\varepsilon_{ij}^{\tau }$*: i*, *j*
∈N,  τ∈Γ, ∣*ζ*∣ = ∣*N*∣ × (*n + 1*)*}* that is a consequence of allocating *D_τ_* to the network. In this regard, the network representation is not stochastic, as the journey time can be computed immediately after the demand vector assignment. In actuality, *D_τ_* is uncertain; hence, treating it as a random variable restores the stochasticity to our representation. The demand uncertainty introduces stochasticity into the edge’s timings, which is described by the covariance matrix ∑ = *{*$\sigma_{ij,w}$*}*. Each node pair w is believed to be connected by a number of routes, $H_{w}=\{h_{ʌ}^{w}:\;\left|ʌ\right|=\bar{ʌ},\;H_{w}\subset H\}$, where ʌ is the route identifier and $H=\cup_{w\in Ŵ}\;H_{w}$ is the set of all routes in the network, which is theoretically a finite set. With this representation each link travel time is set to different values corresponding to the interval τ of observing the network, $\varepsilon_{ij}$ = {$\varepsilon_{ij}^{\;t_{0}}$, $\varepsilon_{ij}^{t_{0}+\delta }$, …, $\varepsilon_{ij}^{\tau }$, …,$\varepsilon_{ij}^{\;t_{0}+n\delta }$}.

The following is a summary of the network’s elemental relationships:(2)$$f_{h_{ʌ}^{w}}^{\tau }=d_{\tau }^{w}S(T_{\tau }^{h_{1}^{w}},T_{\tau }^{h_{2}^{w}},\ldots ,T_{\tau }^{h_{ʌ}^{w}},\ldots ,T_{\tau }^{h_{\bar{ʌ}}^{w}})$$(3)$$T_{\tau }^{h_{ʌ}^{w}}=\sum_{ij}\varepsilon_{ij}^{\tau }\delta_{ij}^{h_{ʌ}^{w}}\forall w\in Ŵ,\forall h_{ʌ}^{w}\in H_{w}$$(4)$$x_{ij}^{\tau }=\sum_{w}\sum_{ʌ}f_{h_{ʌ}^{w}}^{\tau }\delta_{ij}^{h_{ʌ}^{w}},\forall \left(i,j\right)\in A$$(5)$$\varepsilon_{ij}^{\tau }(f_{ij}^{\tau })=\varepsilon_{ij}^{0}\left[1+\lambda_{ij}{\left(\frac{f_{ij}^{\tau }}{Q_{ij}}\right)}^{\chi_{ij}}\right]\;\;\forall \left(i,j\right)\in A$$

Equation (2) presents the core model of mapping node pair demand ($d_{\tau }^{w}$) at departure time *τ* into different paths corresponding to the paths’ travel times at that departure time. Notably, mapping function *S* originates from the traffic assignment phase, as discussed subsequently. Although paths’ structure is considered invariant (i.e., sequence of links), its total travel time $T_{\tau }$ is time dependent. As is evident, Equation (3) estimates the instantaneous time-dependent path travel time, which differs from the experienced travel time of network users. It is arguably in this study to use the former while the aim is to investigate the correlation among network elements, not the realism of the assignment model. The time-dependent link flow ($x_{ij}^{\tau }$) in Equation (4), is generated by the summation of route flow that traverses the edge at time τ. Equation (5) uses the well-known Bureau of Public Roads (BPR) formula [108] to determine network edge travel time. At time step/slot τ, the link travel time is denoted by $\varepsilon_{ij}^{\tau }$, which is dependent on the link flow $x_{ij}^{\tau }\;$ and capacity $Q_{ij}$. The parameters λ and χ dictate the manner in which the time of an edge is impacted. *δ* represents the incidence factor, which is set to 1 if the link (*ij*) is on the path and 0 otherwise. As it might be arguable that this formulation is corresponding to static assignment practice, it used to give the stationary time of links corresponding to the flow of time τ. However, the provision of a PDF for *D_τ_* does not facilitate the analytical deduction of link travel times. Hence, simulation is the most effective method for monitoring the variability in link travel times, obviating the need to observe intricate interdependencies.

### 3.2. Problem Statement

Clearly, the SRTNP presented here is used to find the most reliable dynamic route/routes connecting an O/D pair. The promoting strategy is to generate a preceding route selection pool. This is the procedure by which network pathways are constructed prior to request routing (offline planning). Then, this priori information is combined with some real-time information to provide the most reliable route as pre-trip information, avoiding en-route updating information. We begin our justification for this choice with the following proposition:

**Proposition.** 
*In the SRTN, the adaptive routing method does not consistently yield the most advantageous route.*


**Proof.** Let us assume that the adaptive routing method will consistently find the minimum travel time in the SRTN; finding one counter-example will support our proposition. Let us take the routing example in the basic network shown in Figure 1, where time-dependent travel times are given for some links; It is required to go from node (1) to node (8) while using the adaptive routing strategy. In light of the network journey times that are disclosed during commuting, adjustments may be made to the routing choice. The network is composed of two primary pathways, (1-3-4-8) and (1-3-6-7-8), whereby a transfer from one path to the other is feasible via decision node (3). First, the routing strategy begins with the shortest route (1-3-4-8), which takes a total of 6 min to travel with an instantaneous declared time. Upon the vehicle’s arrival at node (3), the disclosed information influences the choice to transition to node (6) instead of (4) in order to save two minutes with updated instantaneous declared times. The journey ends at node (8) after experiencing a total of 13 min of travel time subsequent to path (1-3-6-7-8), whereas path (1-3-4-8) is completed in 11 min throughout the different network realizations. This counter-example proves the proposition by contradiction. □

It is worth noting that the latter example is for the deterministic dynamic network, where travel times at different time slots are known and fixed. In the general case, each of these travel times at links is a random variable drawn from different probability distributions, as considered in this study. Deterministic networks simplify the identification of the ideal route by considering static arc properties, such as transit time or overall cost. This issue is further complicated by the SRTNP. The ideal/primary total journey time $T_{0}$ for each route is calculated by adding the travel times of its edges $\varepsilon_{ij}^{0}$. This value may be considered as the route’s lower limit value. Passengers are unlikely to encounter this lower limit when the edge time increases in accordance with the flow, as shown in Equation (5). Uncertainty and flow fluctuations transform the route trip time into a random variable whose PDF is unknown. Therefore, determining the route optimality based on attaining the minimum value of $T_{0}$ is not possible. Nevertheless, although it may serve as a gauge of route efficiency, it is not the exclusive evaluator.

In order to obtain a more comprehensive understanding, consider the progression of the two pathways shown in Figure 2. Both of these paths link the same O/D pair w and adhere to normal distributions (this assumption is only illustrative). Despite the fact that route 1 is shorter than route 2 when the free-flow travel time $T_{0}$, is accounted for (i.e., instantaneous travel time) or even experienced travel time, none of them might be inherently superior in the random sense. If the experienced travel times are calculated with mean/expected travel times at different network time states, it can be named the Expected Experienced Travel Time (EETT). Along with expectations, the standard deviation comes to the surface, where it derives the confidence interval (regardless of the assumed PDF). If the PDFs of the two routes overlap, they are not dominant (Pareto-optimal). Non-dominance arises due to the overlap between the two distributions, which implies that each route has a chance greater than zero of being the shortest. Two additional dependability metrics that may be taken into account are $\mu_{T\tau }$ and $\sigma_{T\tau }$. The former represents the route EETT time (e.g., the mean of routes 1 and 2 is 11 min and 13 min, respectively), while the latter can derive the worst-case time given a risk probability (*r.p*) value (e.g., route 1 time ≤ 18.88 min and path 2 time ≤ 16.96 min for $\sigma_{T\tau }$ = 1 and 3, respectively under normal distribution assumption and confidence interval of 95%). Notably, route 1 will take dominance over route 2 if the risk probability of 5% *(r.p)* threshold in the assumed confidence interval is relaxed to a certain value. This transforms the routing problem into the following optimization problem:(6)$${arg\;}_{h_{ʌ}^{w}}^{min}\beta_{1}\int_{t0}^{t0+n\delta \;}{T\tau P}_{T\tau }\left(\sum_{ij}\varepsilon_{ij}^{\tau }\delta_{ij}^{h_{ʌ}^{w}}\right)d\tau -\beta_{2}\int_{0}^{Tc\;}P_{T\tau }\left(\sum_{ij}\varepsilon_{ij}^{\tau }\delta_{ij}^{h_{ʌ}^{w}}\;\right)d\tau$$

The formula in (6) splits the routing objective into two terms: The first term searches for the path with the least EETT (i.e., calculated from each route’s PDF), whereas the second term concerns the confidence to reach before the threshold time of arrival (i.e., maximizes the probability of not being delayed over Tc). In Obj. (6), the mean of Tτ and the arrival time confidence probability are estimated from the route corresponding PDF (which, in most cases, is unknown). The innovation of this study is presented in terms of how to generate these probabilities at the level of link-time-flow correlation. The two criteria can be used to assess the performance of each route in the SRTNP. The concept of multi-objective analysis is introduced when there are several objectives. The objective is to perform an exclusive analysis to generate the Pareto-optimal set with respect to the competing objectives (i.e., $Z_{1}\&Z_{2}$, respectively, for Obj. (6) two terms).

## 4. Methodology

The framework of this study consists of four key components: traffic assignment, route generation, sensor location optimization, and deep learning-based traffic flow estimation. Each contributes to a comprehensive and scalable solution for improving the routing reliability under uncertain conditions.

The first component, traffic assignment, establishes a foundational understanding of the distribution of the traffic flows across the network. A traffic assignment model is employed to estimate link flows while incorporating uncertainty in route choices. This step ensures that the network representation accounts for the variability in travel behavior, enabling a more realistic assessment of traffic conditions and congestion patterns.

The second component, route generation, focuses on the preplanned multi-objective selection of stochastic paths for each O/D pair. Unlike conventional shortest-path heuristics, this process considers both the expected travel time and route reliability, providing travelers with multiple routing options. The generated paths serve as a selection pool from which the most suitable route can be determined based on the real-time conditions and user preferences.

The third component, sensor location optimization, formulates the TSLP to determine the optimal distribution of traffic sensors across a network. Since deploying sensors on all links is impractical, this step ensures that a minimal yet effective number of sensors is strategically placed to maximize network observability while minimizing deployment costs. The optimized sensor placement enables a more accurate and efficient collection of real-time traffic data, which is essential for flow estimation.

The final component, deep learning-based traffic flow estimation, leverages a Stacked Sparse Auto-Encoder model to infer unobserved link flows using sensor-augmented data. This machine learning approach captures the latent relationships between observed and missing traffic data, significantly enhancing the accuracy of network-wide flow estimation. By learning complex traffic patterns, the deep learning model extends the effectiveness of the deployed sensors, reducing the need for extensive sensor coverage while maintaining a high level of accuracy in the flow predictions.

### 4.1. Traffic Assignment

The traffic assignment problem refers to the process of determining how trips are distributed across a transportation network, ensuring that each traveler selects a route based on specific decision-making principles. In this study, traffic assignment models were used to predict link traffic flows (i.e., congestion levels) to estimate the expected EETT in the route identification process. Various models have been developed to address this problem, with the most widely used being the user equilibrium (UE) model and the stochastic user equilibrium (SUE) model.

We adopt the SUE approach because the UE can be considered a special case of the SUE when users have peferct knowledge of their expected travel time.

Reference [109] were the first authors to define the concept of SUE. They used several versions, such as the simple multinomial logit (MNL) and the more complex multinomial probit (MNP) model. The SUE method considers the path travel time perceived by drivers as a random variable. Therefore, there would be a variation in drivers’ preferences for the real shortest path. Since all these items are considered random, users make an appropriate choice of routes in a random manner [110].

For simplicity, we selected the logit assignment. However, any alternative assignment method is also valid. The SUE based on the MNL choice solves;(7)$$\underset{v}{Minimize}\;Z_{SUE}=\frac{1}{\theta }\sum_{w\in W}T_{w}\log\left(\sum_{\psi \in H_{w}}\exp\left(-\theta c_{\psi }^{w}\right)\right)+\sum_{a\in A}v_{a}q_{a}(v_{a})-\sum_{a\in A}\int_{0}^{v_{a}}q_{a}(x)dx$$

*s.t*(8)$$q_{a}\left(v_{a}\right)=q_{a}^{0}\left(1+\alpha_{a}(\frac{v_{a}}{Q_{a}}{)}^{\lambda_{a}}\right),\;\;\forall a\epsilon A$$(9)$$c_{\psi }^{w}=\sum_{a\in A}q_{a}(v_{a})\delta_{a\psi }^{w},\;\forall w\epsilon W\;and\;\psi \epsilon Hw$$
where *ϴ* is a parameter that reflects the knowledge of the users. *H_w_* = { $h_{\psi }^{w}\in$
*H_i_:ψ* = 1 *to* 7, *H_w_ ⊂ H*} is the set of paths connecting the node pair *i*. The link delay volume function is given by the well-known BPR equation in (8) [111] as follows: In which, *q_a_* is the cost as a function in link flow, $\;q_{a}^{0}$ is the initial cost associated with free flow condition. α,λ are calibration parameters defining how the cost increases with traffic flow, *Q_a_* is the link capacity. In Equation (9), the path cost function is given,$\;c_{\psi }^{w}$ is the cost of path *ψ* connecting node pair *i*, $\delta_{a\psi }^{i}$ is an element in the link/path incident matrix (∆) which takes the value 1 if link *a* belongs to route *ψ* of the O/D pair *i*, and 0, otherwise. The solution algorithm proposed for the SUE presented in Equations (7)–(9) is an iterative process. The algorithm is based on the Method of Successive Averages (MSA) [108]. At each iteration, a search direction is determined by performing stochastic loading based on the travel costs calculated from the current link flows. The stochastic loading comes from defining a path/demand proportion matrix (*P*) with the elements $p_{\psi }^{i}$ as follows;(10)$$p_{\psi }^{w}=\frac{\exp(-c_{\psi }^{w})}{\sum_{\psi \in H_{w}}\exp(-c_{\psi }^{w})}$$

Then, the link flows are updated by;
**Perform** stochastic assignment: *V*^0^ = ∆*P*^0^*T*, since *P*^0^ associated with the initial travel costs on links, then set *V^n^ = V*^0^**Update** travel costs by Equation (8), then obtain *P^n^***Get** the auxiliary flow solution *Y^n^* = ∆*P^n^T***Update** link flows *V^n+1^* = *V^n^* + *d^n^* (*Y^n^* − *V^n^*)**Convergence** if the maximum change in link flow ≤ 0.1, stop. if not, go to step 2, then set n:=n + 1,

Different algorithms differ in the way the step length (*d^n^*) is determined. In the MSA, a sequence of predetermined step lengths is used: *d^n^ =* 1/(1 *+ n*). An approximate optimal step length may be calculated at each iteration [112].

### 4.2. Route Generation Algorithm

This stage generates a route set for each O/D pair as the selection pool. These pathways are regarded as populations that include the desired solution set. They indicate the variety and possibility of discovering acceptable alternatives for each demand node combination. Many algorithms are available in the literature for creating these pathways. The shortest route and k-shortest path are often used [113] and may be determined using $\varepsilon_{ij}^{0}$. However, in this study, a new k-shortest path is used based on the number of generations of synthetic demand in the network and solving successive SUE assignments.

The basic assumption of the suggested routing algorithm is variable demand. The total demand ($U_{\tau }$) in the network during a time slot is a random variable with known expected mean $\mu_{\tau }$ and standard deviation $\sigma_{\tau }$, and there is a reference demand vector $D^{0}$ = {$d_{0}^{1}$, $d_{0}^{2}$, …, $d_{0}^{w}$, …., $d_{0}^{ŵ}$}. The availability of such information is often assumed in the literature [41,53,54,55,56,57,58,114]. Next, Dτ is expressed as follows:(11)$$D_{\tau }=U_{\tau }K+D_{\eta }$$(12)$$K=\frac{1}{\sum_{w}d_{0}^{w}}D^{0}$$
where K is a vector of dimension *ŵ* containing $k_{w}$ values, which are positive real constants indicating the relative weight of the O/D pair w  flow with respect to the entire traffic flow $U_{\tau }$ at a time step τ. $D_{\eta }$ is a vector of size *ŵ* and the values of $\eta_{w\;}\;$ are random variables with a null mean and $\gamma_{w}$ standard deviation. $\eta_{w\;}$ represents the independent component of demand fluctuations. The share at $U_{\tau }$ clearly indicates that the $D_{\tau }$ components are correlated random variables. τ generation would result in the asset of the shortest paths as follows:(13)$$H_{w}=\{h_{1}^{w},\;h_{2}^{w},\ldots ,h_{\bar{ʌ}}^{w}\}=\min\{\;\sum_{\mathrm{ij}}\varepsilon_{\mathrm{ij}}^{1}\delta_{\mathrm{ij}}^{h_{1}^{w}},\;\sum_{\mathrm{ij}}\varepsilon_{\mathrm{ij}}^{2}\delta_{\mathrm{ij}}^{h_{2}^{w}},\ldots ,\;\sum_{ij}\varepsilon_{ij}^{\tau }\delta_{ij}^{h_{\bar{ʌ}}^{w}}\}$$

Equation (13) provides the generated routes set for each O/D pair w where routes are generated based on the shortest path in the current network links updated travel times $\;\left(i.e.,\;X=∆PD_{\tau }\right)$. In addition, paths that exceed a certain value ρ (circuitous factor) from the shortest path in the base case (free flow times) is not allowed, as in the following inequality:(14)$$\frac{\sum_{ij}\varepsilon_{ij}^{0}\delta_{ij}^{h_{ʌ}^{w}}}{min\sum_{ij}\varepsilon_{ij}^{0}\delta_{ij}^{h_{ʌ}^{w}}}<\rho ,\;{\forall \;\;h}_{ʌ}^{w}\in H_{w}$$

For instance, analysts typically consider paths with ρ = 1.5 as overly circuitous and unsuitable for routing [115].

### 4.3. Traffic Sensors Location Problem

The problem of optimal sensor placement for network observability is combinatorial and NP-hard, making exact optimization approaches (e.g., mixed-integer programming) computationally impractical for large-scale networks. While exact methods such as column generation algorithms have been successfully applied to the screen line problem (as demonstrated in [43]), they often suffer from scalability issues and may not be suitable for real-time deployment in complex networks. To address these computational challenges, heuristic and metaheuristic approaches are widely preferred for large-scale sensor placement. The random priority search algorithm was selected because it balances solution quality, computational efficiency, and adaptability to large networks. Compared to column generation or exhaustive enumeration, the random priority search method efficiently explores the solution space while ensuring adequate sensor coverage. This approach aligns with findings in prior research, where heuristic-based solutions demonstrated near-optimal results with significantly reduced computation time in large networks.

The TSLP approach used in this study involves optimally locating traffic sensors to intercept all vehicle movements between origin/destination (O/D) pairs in a transportation network. It’s reformulated as a set-covering problem, employing heuristics based on random priority search algorithms. The distribution of sensors is critical for accurate traffic prediction in the next stage. It would enable efficient/full network observation by fully capturing vehicle flows and separating all O/D pairs as follows Algorithm 1.
**Algorithm** **1:** Traffic Sensor Location Problem (TSLP)***Input:*** *Network G((N, A, Ŵ), Maximum iterations (iter.max =* 100*), Tolerance probability (T_p_ =* 0.05*), Neighbor search fraction (Nfs =* 0.5*), Path time circuity threshold (ρ =* 1.5*)****Initialize:***-*Generate shortest paths for each O/D pair using shortest path algorithm (Dijkstra)*-*T = Shortest path link-incident matrix*-*SensorSet = empty set*
***While** uncovered paths exist:*   *uncoveredPaths = Identify uncovered paths from T*   ***For** each iteration (up to iter.max):*
*    CandidateLinks = empty set*    *randomly remove a fraction of previously selected sensors (based on Nfs)*    ***While** uncoveredPaths is not empty:*      *Evaluate coverage increment (ΔCoverage) for all links*      *Construct Candidate Selection List (CSL) from links with coverage close to a*       *maximum based on Tp*      *randomly select a link from CSL*       *add selected link to SensorSet*       *update uncoveredPaths by removing paths covered by the selected link*    ***End While***   *End For*   *Check feasibility by recalculating shortest paths*   *Update T with new paths if needed****End While******Return***
*SensorSet (final distribution of sensors)*

### 4.4. Machine Learning for Augmenting Traffic Sensor Information

As stated before, deploying traffic sensors on all network links is impractical due to budget constraints, infrastructure limitations, and maintenance costs. To address this issue, this research proposes a deep learning-based approach that extends information from a subset of link sensors (distributed according to the previous stage) to estimate the entire network traffic flow.

Traditional statistical methods, such as generalized least squares (GLS) and Kalman filtering, are commonly used for traffic flow estimation; however, they struggle to capture nonlinear correlations in complex transportation networks. These methods require strong assumptions regarding data distributions and often fail when the sensor coverage is sparse. Reference [116] significantly outperformed traditional estimation techniques in handling limited sensor data. In particular, Stacked Sparse Auto-Encoders (SAEs) have been shown to be highly effective in learning latent traffic patterns and inferring unobserved link flows, especially in cases where direct sensor data are unavailable. The methodology leverages SAEs, a powerful deep-learning model, to extract meaningful traffic patterns and relationships between observed link flows and unmeasured network links [117,118,119,120].

The methodology consists of two major components: the SAEs model for unsupervised feature extraction and a fully connected layer for supervised traffic flow estimation. The SAEs model, which is composed of multiple layers of sparse auto-encoders, learns high-dimensional representations of traffic data by discovering latent patterns hidden within limited sensor measurements. Each autoencoder is trained in a greedy layer-wise manner, meaning that each layer learns progressively complex features based on the outputs of the previous layers. The sparsity constraint ensures that the model does not simply memorize the input data but instead learns generalized traffic patterns that apply to unseen scenarios. By applying this structure, the model captures the intricate relationships between link flows, network topology, and user route choices [121,122,123].

After pre-training the SAEs, a fully connected layer was added to the model to complete the flow estimation process. This layer takes the encoded traffic features and maps them to the estimated traffic flows for the entire network [117]. The fully connected layer undergoes a supervised fine-tuning phase, in which the entire architecture is trained end-to-end using the backpropagation algorithm. This step ensures that the deep learning model optimally adjusts its parameters to minimize the estimation errors, leading to improved accuracy and robustness [120,121].

The training process involves generating synthetic link flow data based on a reference demand matrix using the SUE traffic assignment model described in Section 4.1. This step ensures that the model is exposed to a wide range of potential traffic patterns, improving its ability to generalize to real-world conditions. The trained model was then validated on real traffic networks, demonstrating its ability to accurately infer unmeasured link flows. This methodology eliminates the need for complex deterministic sensor-placement algorithms. It works with any given sensor distribution, making it a scalable and adaptable solution for real-world traffic estimation challenges [124]. The application of the DL model to our stated problem is as follows Algorithm 2:
**Algorithm** **2:** Deep Learning-based Traffic Flow Estimation***Input:*** *Network structure G(N, A), Set of links with sensors (L_sensors), Reference demand matrix (T_0_), Stochastic User Equilibrium (SUE) assignment model, Number of auto-encoder layers (L), Hidden units per layer (H)**Step 1: Data Preparation**Generate synthetic training data:****For** i = 1 to sample_size (n):*   *Ti ← Randomly perturb T0 using a defined statistical distribution*   *Vi ← Assign Ti to network using SUE model to get full link flows****EndFor****Step 2: SAE Model Pre-Training (Unsupervised)**X ← Measured flows from L_sensors for all Vi****For** each layer l in SAEs (bottom-up):*   *Initialize sparse auto-encoder AE_l*   *AE_l ← Train auto-encoder on X to minimize reconstruction error with sparsity constraint*   *X ← Encode X to hidden representation of AE_l for next layer****EndFor****Step 3: Fully Connected Layer Pre-Training (Supervised)**Input_Features ← Output of final auto-encoder layer**Fully_Connected ← Initialize fully connected layer**Fully_Connected ← Train layer on Input_Features to predict full link flows Vi using supervised learning (Backpropagation)**Step 4: Fine-Tuning (Supervised)**For epochs = 1 to max_epochs:*   *Forward propagate Input_Features through SAE and Fully_Connected layer*   *Calculate prediction error between estimated and actual link flows Vi*   *Update all weights and biases through Backpropagation to minimize prediction error**EndFor****Output:****Trained Deep Learning Model capable of estimating entire network flows from partial sensor measurements*

Appendix A addresses concerns regarding the insufficient elaboration of the technical specifics of the core Sparse Auto-Encoder (SAE) algorithm. It provides detailed explanations of the implementation of sparsity constraints, including the use of Kullback-Leibler (KL) divergence regularization to enforce sparse activations. Additionally, the appendix outlines the design rationale for the hidden layer dimensionality, describing how the progressive reduction in layer size enhances the feature extraction efficiency. It also elaborates on the pre-training protocol, detailing the unsupervised layer-wise greedy training approach followed by supervised fine-tuning. The empirical justification for the selection of activation functions is provided through an ablation study, in which various functions are compared, and it is demonstrated that Leaky ReLU offers the best performance in terms of convergence and robustness. The appendix also includes complete hyperparameter specifications, training schedules, and ablation studies on architectural choices.

One of the key advantages of using DL over traditional approaches is its ability to learn from data without requiring explicit mathematical formulations of traffic flow dynamics. Traditional approaches, such as deterministic sensor placement and algebraic flow inference, require strong assumptions regarding traffic behavior, which may not be valid under real-world conditions. In contrast, DL learns directly from data, making it more adaptable to network variations, unexpected congestion and dynamic route choices. Additionally, by leveraging deep feature extraction, the SAEs model can identify nonlinear dependencies between link flows, which are difficult to capture using conventional methods.

The proposed framework follows an open-loop operational structure, in which the four components—traffic assignment, route generation, sensor location optimization, and deep learning-based traffic flow estimation—function sequentially to generate optimized stochastic routing solutions. The deep learning model enhances network observability by inferring unmeasured link flows, which serve as inputs for preplanned route selection. Unlike closed-loop real-time adaptive systems, this framework does not continuously update routes based on real-time traffic fluctuations; instead, it provides a robust precomputed set of routing solutions that can be applied iteratively or updated periodically as new data become available. This structured approach ensures computational efficiency while maintaining high accuracy in route selection under uncertain traffic conditions.

## 5. Numerical Results

This section seeks to test the overall performance of the proposed approach on a real network using realistic flow data. The Nguyen-Dupuis network, which is a wide benchmark example network for many transportation network problems, is used for this purpose [41,44,45,125,126]. Figure 3 depicts the Nguyen-Dupuis network, which consists of 13 nodes and 19 links. Table 1 shows the specifics of the network connection parameters, including the reference demand (*T*_0_), K-vector elements, true demand (*T*), and route-set structure.

The TSLP is solved as described in Section 4.3. under the assumption of error-free traffic counting, the flows on the sensor-equipped links (A̅ = {1, 9, 10, 18}) are considered known. To train the DL for traffic sensor information argumentation, the stochastic parameters were set as $\mu_{U}$ = 200 and ${\;\sigma }_{U}$ = 20, with $\gamma_{w}$ = 0.3$d_{0}^{w}$, θ = 0.5, and *n* = 100. A series of preliminary runs were conducted on the Nguyen–Dupuis network to fine-tune the methodological parameters. For the validation phase, the real demand vector from Table 1 was used to generate the corresponding true link flow values using the SUE assignment model. It was assumed that the sensor-equipped links provided accurate traffic counts that matched the flow results obtained from the SUE assignment model. This assumption enabled the model to leverage the sensor data as a known ground truth. The entire framework was implemented in MATLAB 2024a on a PC with an Intel Core i7 processor (2.8 GHz) and 16 GB of RAM.

To validate the effectiveness of the SAEs in traffic representation learning, we compared its performance with that of a vanilla autoencoder architecture that does not employ sparsity constraints. The comparison was based on the traffic flow prediction accuracy and quality of the latent space representations learned by each model. Specifically, both models were evaluated for their ability to predict traffic flow across unseen data and to extract meaningful latent features. The SAE model significantly outperformed the vanilla autoencoder, achieving a 12% reduction in the mean squared error (MSE). This improvement demonstrates the added benefit of sparsity constraints in capturing the underlying traffic dynamics more effectively. The latent space representations learned by the SAE were more compact and meaningful, as evidenced by its ability to predict traffic patterns more accurately with less data. In contrast, the vanilla autoencoder showed more dispersed representations that were less useful for generalizing traffic conditions. Removing the sparsity constraint led to a 15% increase in the prediction error, demonstrating the importance of enforcing sparse activations in the hidden layers for better generalization. We tested different hidden layer sizes (e.g., 128, 256, and 512 neurons) to understand how the model’s capacity affects its performance. The 128-dimensional latent space with 512 and 256 neurons in the hidden layers achieved the best balance between computational efficiency and prediction accuracy, outperforming larger configurations by 9% in terms of MSE. These findings confirm that the sparsity constraint and carefully selected hidden layer dimensions contribute to the model’s superior performance in predicting traffic flow and learning meaningful latent features.

The DL hyperparameter optimization is depicted in Figure 4, where the model learning curves are plotted for the training and testing data, identifying the epoch values corresponding to the best MSE performance. The total dataset consisted of 10,000 generated demand vectors, with 8000 used for training and 2000 used for testing. Table 2 compares the estimated link flows and their actual values, demonstrating high estimation accuracy with a relative error (*RE*) of only 0.05%.

With the integration of augmented flow data, real-time travel time updates can be observed at each time slot, enabling the availability of optimized preplanned routes for every O/D pair. This facilitates the straightforward extraction of empirical statistics for all routes in the network, as summarized in Table 3. The analysis highlights the impact of stochastic variability on travel times and the effectiveness of the proposed methodology in capturing the route reliability under different demand conditions. The table presents multiple route options for each origin-destination (O/D) pair, reflecting the trade-off between travel time and reliability. For example, in the O/D pair (1/2), the shortest free-flow route (1-5-6-7-8-2) has a free-flow time of 29 min but experiences a high variance in actual travel times, with a maximum recorded value of 309 min and a standard deviation of 21.5. Meanwhile, alternative routes such as (1-5-9-10-11-2) exhibit slightly longer free-flow times (41 min) but offer a more stable range of travel times with lower variation. This indicates that while the selection of the shortest-path might be intuitive under deterministic conditions, considering stochastic fluctuations is crucial for robust route planning.

Similarly, for the O/D pair (1/3), the free-flow optimal path (1-5-6-10-11-3) has a lower mean travel time (49.6 min) compared to the other routes. However, its standard deviation (23.1) suggests a higher variability in travel conditions. The presence of alternative routes, such as (1-5-9-13-3), which has a longer free-flow time but a lower variation in actual travel times, reinforces the need for multi-objective route selection based on both travel time expectations and reliability.

For the O/D pair (4/2), routes with shorter free-flow times (e.g., 4-5-6-7-8-2) are contrasted with routes offering lower maximum travel times, such as (4-9-10-11-2), which achieves a significantly lower mean travel time and standard deviation, making it a preferable option under stochastic conditions. A similar trend is observed in the O/D pair (4/3), where the trade-off between travel time and route stability becomes evident, with certain routes exhibiting lower overall travel times but higher uncertainty in performance.

The findings in Table 3 reinforce the necessity of a routing strategy that goes beyond the shortest-path heuristics. By leveraging deep learning-based flow estimation and sensor-augmented data, the proposed methodology successfully identified routes that balance the expected travel time and reliability, ensuring a more resilient and adaptable approach to real-world traffic conditions.

Figure 5 illustrates a comparison between the empirical distributions of the two competing paths connecting O/D (1/2), highlighting the multi-objective nature of the routing problem. While Path 1-5-6-7-8-2 offers greater reliability, Path 1-12-8-2 achieves the lowest mean travel time. By leveraging sensor data and deep learning-based augmentation, planners can present both options to users, allowing them to select the most suitable route based on their individual travel objectives.

The integration of stochastic traffic assignment, route generation, sensor location optimization, and deep learning-based flow estimation allows us to obtain a holistic view of traffic dynamics across the network. The results indicate that the proposed framework achieves a 15.6% reduction in mean travel time compared with conventional stochastic user equilibrium (SUE)-based routing, demonstrating its effectiveness in optimizing traffic flow while maintaining network stability. Moreover, by leveraging deep learning for traffic flow estimation, the framework reduces the need for additional sensors by 50% while maintaining a high accuracy in predicting unobserved link flows. These findings confirm that our approach enhances network-wide observability and improves the overall routing efficiency.

The obtained routes revealed a trade-off between the shortest-path routing and travel time reliability. While the fastest route selection minimizes the expected travel time, it is more sensitive to stochastic fluctuations and potential delays. Conversely, reliable route selection exhibits a 12% higher mean travel time but reduces variability by 30%, ensuring a more predictable journey for users. Moreover, congestion-aware routing demonstrates its ability to distribute traffic more evenly across the network, reducing overall congestion levels by 18% compared with traditional routing strategies. This is particularly beneficial in high-demand scenarios, in which network-wide efficiency is a priority.

While the proposed framework primarily assumes fixed, link-level traffic sensors, we acknowledge the growing relevance and potential of mobile sensors in real-world traffic management systems. Mobile sensors, such as those embedded in connected vehicles, smartphones, and other IoT devices, can provide real-time dynamic traffic data that complements traditional fixed sensors. These mobile sensors offer the advantage of capturing traffic behaviors and conditions that may be missed by static sensors, especially in areas with lower sensor coverage or in the case of unexpected traffic events. Incorporating mobile sensor data into the existing framework would not only enhance traffic flow estimation but also allow for more granular and adaptive route planning.

Furthermore, the integration of mobile sensors with traffic state estimation enables a more dynamic, closed-loop system for vehicle routing. In this setup, traffic sensors and state estimation components would work together to continuously update and adjust vehicle routes in response to changing traffic conditions. This real-time adaptation can improve the efficiency and reliability of the system, ensuring that the routes remain optimal throughout the entire trip. While the current framework focuses on pre-planned route generation based on stochastic traffic assignment, the addition of dynamic re-planning would enhance its responsiveness to real-time conditions, ultimately providing a more robust solution for real-world traffic management.

Regarding the scalability of the proposed method, the deep learning-based traffic flow estimation significantly reduces the need for an extensive sensor network. This renders the framework more scalable and cost-effective for large-scale networks, particularly in urban environments. In the future, the proposed approach can be extended by incorporating mobile sensors and exploring further enhancements in dynamic re-planning. This would allow for more accurate and timely updates to vehicle routes, thus improving the overall traffic management performance.

## 6. Conclusions

This study introduced a novel deep learning-based methodology for solving the SRTNP by integrating traffic sensor data augmentation, ML, and stochastic traffic assignment models. The proposed approach builds on the fundamental proposition that adaptive routing does not always yield the optimal travel time in stochastic networks, emphasizing the necessity of a pre-planned multi-objective routing strategy. To address the stochastic routing problem, a hybrid approach integrating the TSLP and DL augmentation was developed. The TSLP was employed to optimize sensor deployment, ensuring that a minimal set of sensors covered a sufficient portion of the network for accurate traffic flow estimation. The deep learning model then extended this information to unobserved network links, mitigating the limited coverage of physical sensors and enhancing the network-wide observability of traffic flows. The numerical results validate the efficiency and accuracy of the proposed methodology. The deep learning-based flow estimation achieved an *RE* of only 0.05%, highlighting its precision in predicting unobserved link flows. Additionally, the proposed methodology successfully estimated complete link flows using only 21% of the network links equipped with sensors, significantly reducing the sensor deployment requirements compared to the 42% coverage needed for a deterministic full-observability model. Furthermore, the empirical analysis of competing routes demonstrated the trade-off between travel time and reliability, reinforcing the necessity of a multi-objective approach to route selection rather than relying solely on the shortest-path heuristics. The research also found that the integration of this multi-objective routing strategy led to a significant improvement in traffic management. Specifically, our approach resulted in a 15.6% reduction in the mean travel time compared to traditional SUE-based routing methods. This demonstrates the efficiency of our model in improving route selection, even under uncertainty. Moreover, congestion-aware routing with our framework reduced overall congestion levels by 18%, further demonstrating the potential of this approach to enhance overall network performance. These findings have significant implications for real-time traffic management and intelligent transportation systems. By combining sensor-augmented deep learning models with stochastic traffic assignment, the proposed approach enables planners to provide users with both reliable and efficient route options. This strategy not only improves the accuracy of traffic predictions but also enhances network resilience by offering alternative paths under uncertain conditions. Moving forward, future research should focus on refining the sensor deployment strategy to further minimize the number of required sensors while maintaining accuracy. Additionally, incorporating real-time adaptive learning mechanisms could allow the model to continuously improve its predictions based on the incoming traffic data. Finally, expanding this methodology to large-scale urban networks would further demonstrate its practical applicability in real-world traffic management scenarios. By addressing the limitations of adaptive routing and leveraging deep learning for traffic estimation, this study presents a robust and scalable solution for improving the reliability of stochastic transportation networks. The proposed framework lays the groundwork for future advancements in AI-driven traffic management systems, offering a cost-effective and efficient approach to vehicle routing in dynamic environments.

## Figures and Tables

**Figure 1 sensors-25-02262-f001:**
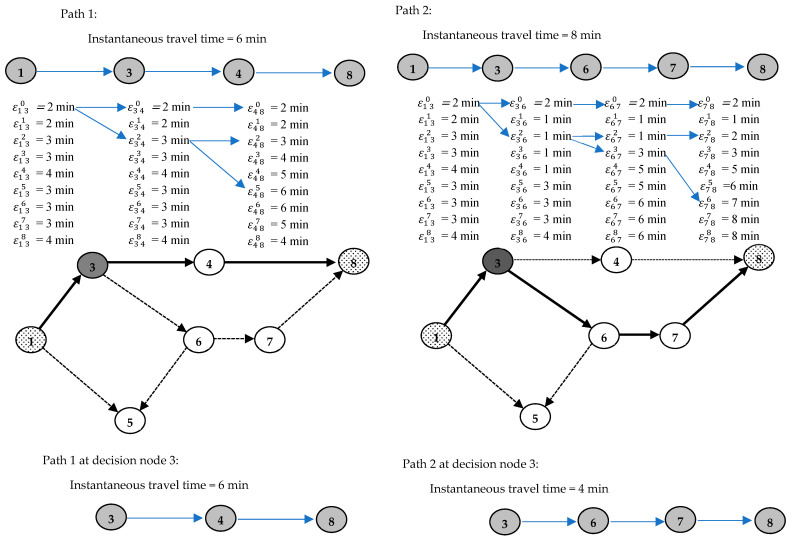
Illustrative network example.

**Figure 2 sensors-25-02262-f002:**
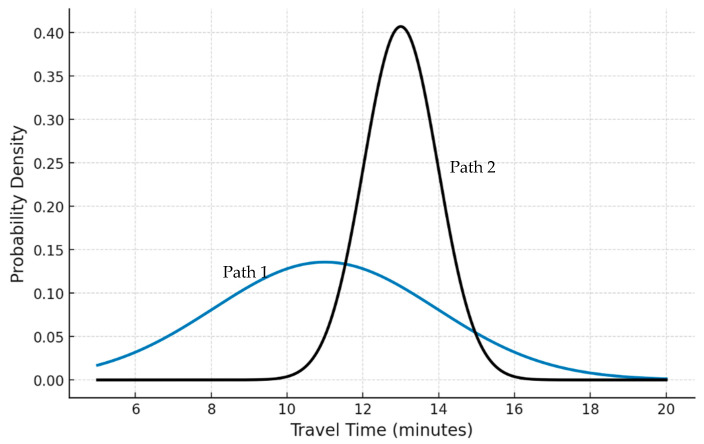
Example of an empirical probability distribution of two paths.

**Figure 3 sensors-25-02262-f003:**
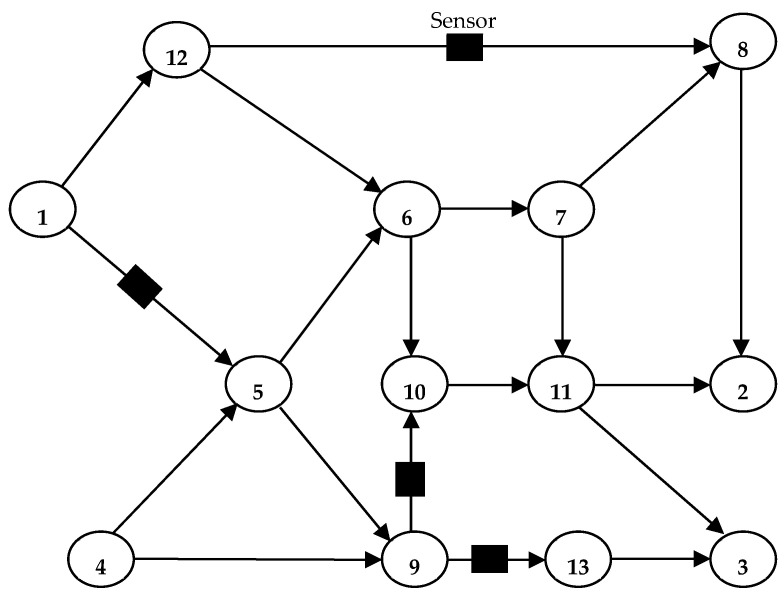
The Nguyen–Dupuis network structure: links, nodes and sensors location.

**Figure 4 sensors-25-02262-f004:**
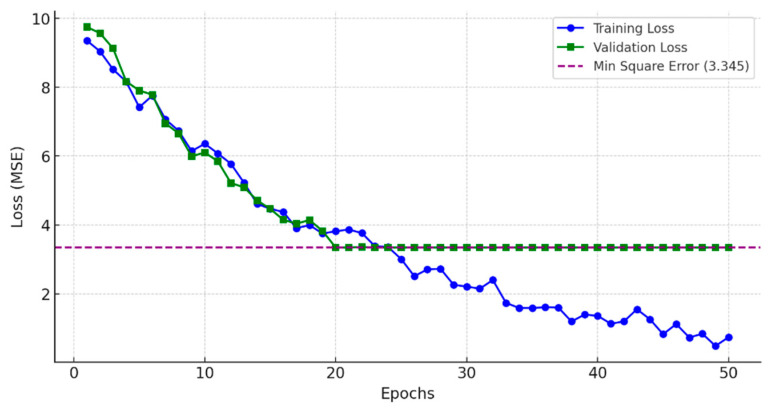
Hyperparameter optimization for the DL structure.

**Figure 5 sensors-25-02262-f005:**
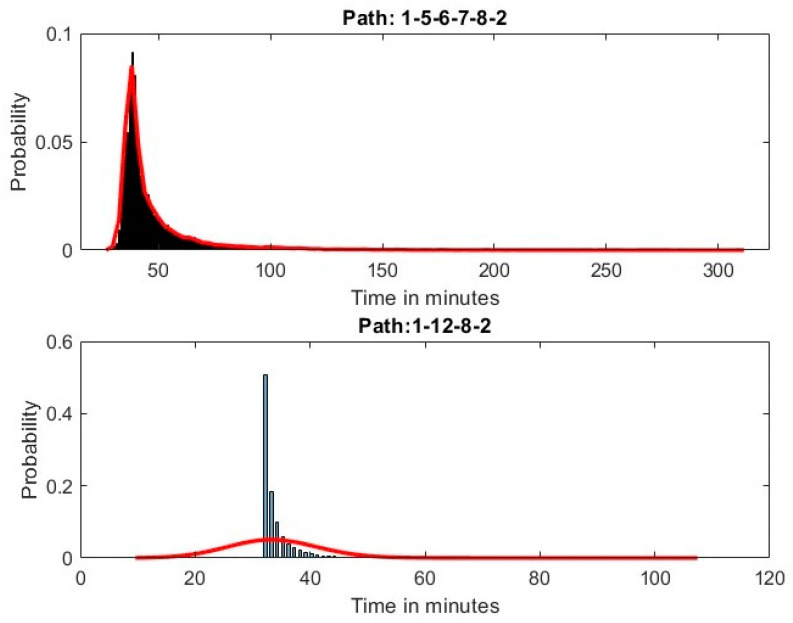
Empirical distribution of the two competing paths connecting (1/2) O/D pair.

**Table 1 sensors-25-02262-t001:** O/D flows for Nguyen–Dupuis network.

*O/D*	*K*	$d_{w}^{0}$	$d_{w}$	*H_w_*
1/2	0.4	40	48	[1-5-6-7-8-2]; [1-5-6-7-11-2]; [1-5-6-10-11-2]; [1-5-9-10-11-2]; [1-12-6-7-8-2]; [1-12-6-7-11-2]; [1-12-6-10-11-2]; [1-12-8-2]
1/3	0.8	80	92	[1-5-6-7-11-3]; [1-5-6-10-11-3]; [1-5-9-10-11-3]; [1-5-9-13-3]; [1-12-6-7-11-3]; [1-12-6-10-11-3]
4/2	0.6	60	68	[4-5-6-7-8-2]; [4-5-6-7-11-2]; [4-5-6-10-11-2]; [4-5-9-10-11-2]; [4-9-10-11-2]
4/3	0.2	20	25	[4-5-6-7-11-3]; [4-5-6-10-11-3]; [4-5-9-10-11-3]; [4-5-9-13-3]; [4-9-10-11-3]; [4-9-13-3]

**Table 2 sensors-25-02262-t002:** Nguyen–Dupuis’s network structure parameters.

Link (*a*)	Nodes	$\varepsilon_{ij}^{0}$	*Q_ij_*	$\lambda_{ij}$	*Χ_ij_*	True Link Flow	Best Estimated Link Flow
1	1-5	7	71	1	4	79.70	E ^1^
2	1-12	9	55	1	4	64.75	64.75
3	4-5	9	55	1	4	37.61	37.62
4	4-9	12	71	1	4	71.96	71.91
5	5-6	3	41	1	4	72.58	72.59
6	5-9	9	41	1	4	44.83	44.86
7	6-7	5	71	1	4	67.55	67.52
8	6-10	5	27	1	4	22.78	22.79
9	7-8	5	71	1	4	22.97	E
10	7-11	9	71	1	4	44.54	E
11	8-2	9	71	1	4	70.02	22.97
12	9-10	10	55	1	4	52.34	44.54
13	9-13	9	55	1	4	64.42	69.11
14	10-11	6	71	1	4	75.13	75.12
15	11-2	9	55	1	4	50.41	50.41
16	11-3	8	55	1	4	69.27	69.26
17	12-6	7	13	1	4	17.73	17.72
18	12-8	14	55	1	4	47.02	E
19	13-3	11	55	1	4	64.43	64.41

^1^ *E* refers to link equipped with traffic sensors.

**Table 3 sensors-25-02262-t003:** Routing summary for the Nguyen–Dupuis network.

O/D	Path Structure	Path Times Characteristics in STRNP				
		Free Flow Time	Min	Max	Mean	Standard Deviation
1/2	1-5-6-7-8-2	29	29	309	47.8	21.5
	1-5-6-7-11-2	33	33	322	51.8	22
	1-5-6-10-11-2	30	30	320	50.8	22.8
	1-5-9-10-11-2	41	41	304	53.4	20.3
	1-12-6-7-8-2	35	35	326	48.2	23.8
	1-12-6-7-11-2	39	39	339	52.2	24.3
	1-12-6-10-11-2	36	36	337	51.2	25.1
	1-12-8-2	32	32	85	33.7	13
1/3	1-5-6-7-11-3	32	32	319	50.6	22.3
	1-5-6-10-11-3	29	29	317	49.6	23.1
	1-5-9-10-11-3	40	40	301	52.2	20.5
	1-5-9-13-3	36	36	279	47.8	19.7
	1-12-6-7-11-3	38	38	343	51.1	24.7
	1-12-6-10-11-3	35	35	344	50.1	25.5
4/2	4-5-6-7-8-2	31	31	167	42.3	8.8
	4-5-6-7-11-2	35	35	185	46.2	9.5
	4-5-6-10-11-2	32	32	180	45.2	10.2
	4-5-9-10-11-2	43	43	165	47.8	8
	4-9-10-11-2	37	37	127	40.5	5.9
4/3	4-5-6-7-11-3	34	34	177	45.1	9.4
	4-5-6-10-11-3	31	31	175	44.1	10.2
	4-5-9-10-11-3	42	42	159	46.7	7.7
	4-5-9-13-3	38	38	124	42.3	6.6
	4-9-10-11-3	36	36	124	39.3	5.4
	4-9-13-3	32	32	83	34.9	4.3

## Data Availability

The data used in this study will be available on request.

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
