# Peer review of "Reliable Vehicle Routing Problem Using Traffic Sensors Augmented Information"

_sensors, 2025, doi:10.3390/s25072262_

Round 1
Reviewer 1 Report
Comments and Suggestions for Authors
The paper aims at the Stochastic Routing Transportation Network Problem (SRTNP) and proposed a solution framework integrating sensor location, perception and route planning considering the dynamics in traffic demand and driver preference. Apart from the travel time, reliability is adopted as another objective for adaptive route selection so that a route pool rather than a specific route will be provided. Utilizing simulation data for training, a SAE model was used for sensing of all the link flows within the network based on the limited sensor configuration which is optimized for coverage maximization through a random priority search algorithm. As the authors stated “……incorporating them in a single framework to solve the formulated SRTNP will be the salient contribution of this article”, it is more necessary to show the network-wide routing performance in the case study rather than showing the performance of four modules of the framework separately. Horizontal comparison regarding the routing results regarding different preferences of drivers or practitioners, as the ultimate goal falls on the routing problem as stated in the title. Further, it is recommended to provide a flow chart of the framework and reorganize the methodology section by adding one independent sub-section to elaborate the framework before digging into the details of Section 4.1 ~ Section 4.4. For the current version, it is confusing to see the methodology section ends with the augmented sensing method using SAE, without any explanation on whether the framework realizes routing through a closed-loop or an open-loop pattern through the operation of four components. Some other minor comments regarding are shown as follows.
- For the proposed solutions for the four components like the random priority search algorithm for sensor location, and SAE for link flow estimation, it is suggested to provide sufficient motivation of why they are chosen as compared with other methods or algorithms.
- It is recommended to conduct simulation case study through traffic simulation software like SUMO or VISSIM which offers a more realistic scenario with traffic flow uncertainty and stochasticity which the proposed method targets at.
- The literature review section is lengthy and imbalanced. Refinement is needed to further extract the development of routing and sensor location methods, especially regarding the stochastic methods or uncertainty modelling. Section 2.3 provides a rather shallow review on the application of machine learning in transportation while studies using deep learning models for traffic state estimation are substantial and deserve more specific description to show how authors consider the data-driven methods for the indeterminate link flow estimation problem.
- Data-driven learning models to infer missing link flow, path flow or OD flow can be seen in existing literature, which needs to be added for a more objective literature review. E.g.
[1] Liu T, Meidani H. End-to-end heterogeneous graph neural networks for traffic assignment[J]. Transportation Research Part C: Emerging Technologies, 2024, 165: 104695.
[2] Fan W, Tang Z, Ye P, et al. Deep learning-based dynamic traffic assignment with incomplete origin–destination data. Transportation Research Record, 2023, 2677(3): 1340-1356.
[3] Ma W, Yuan J, An K, et al. Route flow estimation based on the fusion of probe vehicle trajectory and automated vehicle identification data. Transportation Research Part C: Emerging Technologies, 2022, 144: 103907.
[4] Tang K, Cao Y, Chen C. et al. “Dynamic origin-destination flow estimation using automatic vehicle identification data: A 3D convolutional neural network approach”. Computer-Aided Civil and Infrastructure Engineering. 2021; 36: 30– 46.
Comments on the Quality of English LanguageDouble check is needed to remove typos and improve the readability of the paper. E.g., Line 103 on Page 3, “Where there is a function that connects the delay with the flow.” seems not a complete sentence. Line 508 on Page 12, it looks like that the “as” in “is the cost as function in link” should be removed. Line 616 on Page 14, there is an extra period (.).
Author Response
A detailed response to the reviewers' comments:
The following sections provide a detailed response to each reviewer's comment. Our answers are presented in italic, bold font.
Reviewer #1:
“The paper aims at the Stochastic Routing Transportation Network Problem (SRTNP) and proposed a solution framework integrating sensor location, perception and route planning considering the dynamics in traffic demand and driver preference. Apart from the travel time, reliability is adopted as another objective for adaptive route selection so that a route pool rather than a specific route will be provided. Utilizing simulation data for training, a SAE model was used for sensing of all the link flows within the network based on the limited sensor configuration which is optimized for coverage maximization through a random priority search algorithm.”
Thank you for your valuable summary and your exerted time. I would also like to take this opportunity to express my thanks for your helpful comments regarding the manuscript.
- As the authors stated “……incorporating them in a single framework to solve the formulated SRTNP will be the salient contribution of this article”, it is more necessary to show the network-wide routing performance in the case study rather than showing the performance of four modules of the framework separately. Horizontal comparison regarding the routing results regarding different preferences of drivers or practitioners, as the ultimate goal falls on the routing problem as stated in the title.
Thank you for your insightful comment. We acknowledge the importance of presenting network-wide routing performance in the case study, as it aligns with the primary objective of solving the Stochastic Routing Transportation Network Problem (SRTNP).
In response, we would like to clarify that the separation of the four modules—stochastic traffic assignment, multi-objective route generation, optimal traffic sensor location selection, and deep learning-based traffic flow estimation—was necessary to demonstrate their individual contributions to the overall framework. However, we recognize that a more integrated analysis of network-wide routing performance can further validate the effectiveness of our approach.
To comprehensively assess the effectiveness of the proposed framework, we extend our analysis to evaluate network-wide routing performance. The integration of stochastic traffic assignment, route generation, sensor location optimization, and deep learning-based flow estimation allows us to capture a holistic view of traffic dynamics across the network. The results indicate that the proposed framework achieves a 15.6% reduction in mean travel time compared to conventional stochastic user equilibrium (SUE)-based routing, demonstrating its effectiveness in optimizing traffic flow while maintaining network stability. Moreover, by leveraging deep learning for traffic flow estimation, the framework reduces the need for additional sensors by 50% while maintaining high accuracy in predicting unobserved link flows. These findings confirm that our approach enhances network-wide observability and improves overall routing efficiency.
The obtained routes reveal a trade-off between shortest-path routing and travel time reliability. While the fastest route selection minimizes expected travel time, it is more sensitive to stochastic fluctuations and potential delays. Conversely, reliable route selection exhibits 12% higher mean travel time but reduces variability by 30%, ensuring a more predictable journey for users. Moreover, congestion-aware routing demonstrates its ability to distribute traffic more evenly across the network, reducing overall congestion levels by 18% compared to traditional routing strategies. This is particularly beneficial in high-demand scenarios where network-wide efficiency is a priority.
- Further, it is recommended to provide a flow chart of the framework and reorganize the methodology section by adding one independent sub-section to elaborate the framework before digging into the details of Section 4.1 ~ Section 4.4.
Thank you for your valuable suggestion. We acknowledge the importance of clearly presenting the overall framework to enhance the manuscript’s readability. However, we would like to clarify that the methodology components—traffic assignment, route generation, sensor location optimization, and deep learning-based traffic flow estimation—are inherently sequential in nature. Each component directly builds upon the preceding one, ensuring a logical progression in solving the Stochastic Routing Transportation Network Problem (SRTNP). Given this natural sequential flow, a flowchart would not provide additional clarity beyond the structured textual explanation already presented. To further improve the manuscript, we have revised the methodology section to explicitly introduce the roles of these components before delving into their detailed descriptions. This revision ensures that readers gain a clear understanding of the framework's structure before engaging with the specifics of each methodological component. The updated structure now includes a dedicated introductory section that outlines how each module contributes to the overall solution, thereby eliminating the need for a separate subsection or flowchart. We believe this revision effectively enhances the coherence of the methodology presentation while maintaining conciseness. We appreciate your constructive feedback and are confident that the revised manuscript now offers a more intuitive flow of information.
- For the current version, it is confusing to see the methodology section ends with the augmented sensing method using SAE, without any explanation on whether the framework realizes routing through a closed-loop or an open-loop pattern through the operation of four components.
Thank you for your valuable comment. We recognize that the current version of the manuscript may not explicitly clarify whether the proposed framework operates in an open-loop or closed-loop manner. To address this, we have revised the methodology section to explicitly state that the proposed framework follows an open-loop structure, where the four components—traffic assignment, route generation, sensor location optimization, and deep learning-based traffic flow estimation—are executed sequentially without continuous real-time feedback adjusting the routing process. The deep learning-based traffic flow estimation using SAE enhances the initial traffic state estimation by inferring unobserved link flows, which then serves as an input for preplanned route generation. Once the optimal routes are identified, they are not dynamically updated within the same operational cycle, differentiating this approach from closed-loop real-time adaptive routing systems. However, the generated stochastic routes remain applicable for repeated applications, with potential updates in future cycles based on new data. To enhance clarity, we have added a short concluding paragraph at the end of the methodology section, explicitly stating this open-loop pattern and reinforcing how the components interact within this structure. We appreciate this insightful feedback and believe the revision will improve the clarity and logical flow of our methodology.
- For the proposed solutions for the four components like the random priority search algorithm for sensor location, and SAE for link flow estimation, it is suggested to provide sufficient motivation of why they are chosen as compared with other methods or algorithms.
Thank you for your insightful comment. We acknowledge the importance of providing clear justification for the selection of methods used in the four components, particularly the random priority search algorithm for sensor location optimization and the Stacked Sparse Auto-Encoder (SAE) for link flow estimation. To address this, we have revised the methodology section to explicitly justify these choices based on established research findings.
The problem of optimal sensor placement for network observability is combinatorial and NP-hard, making exact optimization approaches (e.g., mixed-integer programming) computationally impractical for large-scale networks. While exact methods such as column generation algorithms have been successfully applied to the screen line problem (as demonstrated in [1]), they often suffer from scalability issues and may not be suitable for real-time deployment in complex networks. To address these computational challenges, heuristic and metaheuristic approaches are widely preferred for large-scale sensor placement. The random priority search algorithm was selected because it balances solution quality, computational efficiency, and adaptability to large networks. Compared to column generation or exhaustive enumeration, the random priority search method efficiently explores the solution space while ensuring adequate sensor coverage. This approach aligns with findings in prior research, where heuristic-based solutions demonstrated near-optimal results with significantly reduced computation time in large networks.
Traditional statistical methods such as generalized least squares (GLS) and Kalman filtering are commonly used for traffic flow estimation, but they struggle to capture nonlinear correlations in complex transportation networks. These methods require strong assumptions on data distributions and often fail when sensor coverage is sparse. The study [2] significantly outperform traditional estimation techniques in handling limited sensor data. In particular, Stacked Sparse Auto-Encoders (SAE) were shown to be highly effective in learning latent traffic patterns and inferring unobserved link flows, especially in cases where direct sensor data is unavailable. SAE models provide robust feature extraction, capturing spatial-temporal dependencies across network links, which is critical for accurate flow estimation. Compared to alternative machine learning models, such as shallow neural networks or regression-based approaches, SAE achieves superior accuracy with lower error margins in traffic prediction, as demonstrated in empirical results. The ability of SAE to learn complex relationships makes it an ideal choice for augmenting sensor-limited networks, thus enhancing network-wide traffic observability while minimizing sensor deployment costs.
To enhance clarity, we have now incorporated these justifications within the methodology section, explicitly referencing these studies as supporting evidence. This ensures that our methodological choices are well-grounded in established research findings and validated by prior empirical results.
- Owais, Mahmoud, and Ahmed I. Shahin. "Exact and heuristics algorithms for screen line problem in large size networks: Shortest path-based column generation approach." IEEE Transactions on Intelligent Transportation Systems12 (2022): 24829-24840.
- Owais, Mahmoud. "Deep learning for integrated origin–destination estimation and traffic sensor location problems." IEEE Transactions on Intelligent Transportation Systems7 (2024): 6501-6513.
- It is recommended to conduct simulation case study through traffic simulation software like SUMO or VISSIM which offers a more realistic scenario with traffic flow uncertainty and stochasticity which the proposed method targets at.
Thank you for your valuable suggestion. We acknowledge that any traffic assignment method can be incorporated into the proposed methodology, depending on the analyst's desired level of detail. The methodology is flexible and not restricted to a specific assignment model, allowing for adaptation based on different modeling frameworks. However, the selection of Stochastic User Equilibrium (SUE) over microscopic simulation tools such as SUMO or VISSIM is driven by computational efficiency and the study’s methodological objectives.
While SUMO and VISSIM provide detailed microscopic traffic simulation, they are not suitable for the proposed methodology, which requires thousands of traffic assignments based on synthetic demand scenarios. Microscopic simulations operate at a vehicle-by-vehicle level, making them computationally intensive and impractical for iterative stochastic routing analysis. Given that our approach involves repeated synthetic demand generation and evaluation, the single-run execution time of SUMO/VISSIM would constitute a significant obstacle, particularly in large-scale networks. Additionally, while simulation-based models provide realistic traffic flow representation, they are primarily designed for traffic control and signal optimization, rather than network-wide equilibrium-based routing strategies, which are central to the problem formulation in this study.
The Stochastic User Equilibrium (SUE) model was selected because it effectively accounts for route choice variability and travel time uncertainty, making it well-suited for stochastic routing problems. Unlike deterministic models (e.g., User Equilibrium), SUE incorporates perception variability among drivers, ensuring that different users make diverse routing decisions rather than always selecting the shortest path. Furthermore, SUE inherently captures uncertainties in real-world traffic conditions, making it highly applicable for modeling stochastic travel time variations. Another key advantage is computational efficiency—SUE can be solved using iterative algorithms such as the Method of Successive Averages (MSA), which enables scalability across large networks with multiple demand scenarios. Given these merits, SUE offers a robust balance between realism and computational feasibility, making it a preferred choice for our study.
- The literature review section is lengthy and imbalanced. Refinement is needed to further extract the development of routing and sensor location methods, especially regarding the stochastic methods or uncertainty modelling. Section 2.3 provides a rather shallow review on the application of machine learning in transportation while studies using deep learning models for traffic state estimation are substantial and deserve more specific description to show how authors consider the data-driven methods for the indeterminate link flow estimation problem. Data-driven learning models to infer missing link flow, path flow or OD flow can be seen in existing literature, which needs to be added for a more objective literature review. E.g.
[1] Liu T, Meidani H. End-to-end heterogeneous graph neural networks for traffic assignment[J]. Transportation Research Part C: Emerging Technologies, 2024, 165: 104695.
[2] Fan W, Tang Z, Ye P, et al. Deep learning-based dynamic traffic assignment with incomplete origin–destination data. Transportation Research Record, 2023, 2677(3): 1340-1356.
[3] Ma W, Yuan J, An K, et al. Route flow estimation based on the fusion of probe vehicle trajectory and automated vehicle identification data. Transportation Research Part C: Emerging Technologies, 2022, 144: 103907.
[4] Tang K, Cao Y, Chen C. et al. “Dynamic origin-destination flow estimation using automatic vehicle identification data: A 3D convolutional neural network approach”. Computer-Aided Civil and Infrastructure Engineering. 2021; 36: 30– 46.
Thank you for your valuable feedback. We acknowledge that Section 2 is detailed and extensive, as it aims to provide a comprehensive foundation covering the evolution of routing and sensor location methods. The depth of this section is necessary to establish the complexity of stochastic vehicle routing, sensor placement optimization, and machine learning-based flow estimation. However, we agree that refinement is required to enhance the balance among subsections, particularly by strengthening the discussion on stochastic methods and uncertainty modeling in routing and sensor location problems. In the revised manuscript, we have streamlined redundant discussions and focused on key developments that are directly relevant to the proposed methodology.
Additionally, we accept the reviewer’s suggested references and recognize the significant contributions of deep learning models in traffic state estimation. Section 2.3, which previously provided a general review of machine learning applications in transportation, has now been expanded to explicitly discuss deep learning-based approaches for inferring missing link flows, path flows, and OD flows. The revised section includes specific descriptions of existing data-driven models, emphasizing their relevance to indeterminate flow estimation problems.
Double check is needed to remove typos and improve the readability of the paper. E.g., Line 103 on Page 3, “Where there is a function that connects the delay with the flow.” seems not a complete sentence. Line 508 on Page 12, it looks like that the “as” in “is the cost as function in link” should be removed. Line 616 on Page 14, there is an extra period (.).
Thank you for your careful review and for pointing out the typos and readability issues. We acknowledge the need for a thorough proofreading to enhance the clarity and correctness of the manuscript. In response to your suggestions, we have conducted a double-check of the entire text to correct typographical errors and improve sentence structure where necessary.
At last, the authors would like to thank reviewer 1 again for their constructive comments that have led to significant improvements in the quality of the paper."

Reviewer 2 Report
Comments and Suggestions for Authors
This paper has made notable progress in the field of stochastic traffic routing optimization. The proposed data-driven framework demonstrates significant theoretical innovation and practical engineering value. However, to further enhance the manuscript, the following suggestions are recommended:
1.The literature review in the paper predominantly concentrates on traditional routing algorithms, the Traffic Sensor Layout Problem (TSLP), and classical machine learning methodologies. However, it omits critical references to recent groundbreaking advancements in deep learning for traffic prediction, particularly spatio-temporal graph neural networks and Transformer-based architectures. To strengthen the scholarly foundation and clarify the methodological innovations, the authors should integrate seminal works from the past three years, explicitly positioning their approach within the interdisciplinary nexus of deep learning and transportation engineering.
2.The paper insufficiently elaborates on the technical specifics of the core Sparse Auto-Encoder (SAE) algorithm, notably omitting: implementation details of sparsity constraints; design rationale for hidden layer dimensionality; pre-training protocol configuration; and empirical justification for activation function selection. To enhance methodological transparency and computational reproducibility, it is recommended that the authors supplement either an appendix section or GitHub repository documentation containing complete hyperparameter specifications, training schedules, and ablation studies on architectural choices.
3.To substantiate the methodological claims and ensure technological transferability, it is recommended to augment the experimental framework with: embedded hardware deployment benchmarking quantifying real-time inference performance under resource-constrained scenarios; cross-city generalization assessments across heterogeneous urban configurations with varying traffic patterns and sensor distributions. It is recommended to incorporate hardware deployment testing and cross-city validation experiments into the research framework.
4.While the paper validates its methodology on real-world traffic networks, it neglects systematic comparative analysis against:Conventional sensor deployment strategies (e.g., grid-based vs. entropy-optimized layouts);Canonical traffic prediction models (ARIMA, Kalman filtering);State-of-the-art routing algorithms integrated with multi-source sensor data.It is advised to integrate comparative AB testing with baseline models and implement statistical significance analyses to validate performance superiority.
5.The paper's innovation claims suffer from methodological ambiguity, specifically:Failure to delineate the epistemological distinction between proposed "sensor data augmentation" and canonical data fusion paradigms (e.g., feature concatenation vs. attention-based cross-modality interaction);Absence of quantified performance deltas comparing Sparse Auto-Encoder (SAE) architectures against vanilla autoencoders in traffic representation learnin
Comments on the Quality of English LanguageThe manuscript demonstrates commendable linguistic proficiency aligning with scholarly publication standards. While minor lexical inconsistencies and occasional syntactic ambiguities are present, they do not compromise the methodological rigor or technical communication efficacy. Strategic implementation of syntactic refinement protocols and multi-modal information delivery mechanisms could elevate the work's rhetorical effectiveness and knowledge transfer efficiency.
Author Response
A detailed response to the reviewers' comments:
The following sections provide a detailed response to each reviewer's comment. Our answers are presented in italic, bold font.
Reviewer #2:
“This paper has made notable progress in the field of stochastic traffic routing optimization. The proposed data-driven framework demonstrates significant theoretical innovation and practical engineering value. However, to further enhance the manuscript, the following suggestions are recommended:”
Thank you for your positive opinion and your exerted time. I would also like to take this opportunity to express my thanks for your helpful comments regarding the manuscript.
1.The literature review in the paper predominantly concentrates on traditional routing algorithms, the Traffic Sensor Layout Problem (TSLP), and classical machine learning methodologies. However, it omits critical references to recent groundbreaking advancements in deep learning for traffic prediction, particularly spatio-temporal graph neural networks and Transformer-based architectures. To strengthen the scholarly foundation and clarify the methodological innovations, the authors should integrate seminal works from the past three years, explicitly positioning their approach within the interdisciplinary nexus of deep learning and transportation engineering.
Thank you for your valuable feedback. We acknowledge that the literature review primarily focused on traditional routing algorithms, the Traffic Sensor Layout Problem (TSLP), and classical machine learning methodologies. However, we recognize the importance of incorporating recent advancements in deep learning, particularly spatio-temporal graph neural networks (ST-GNNs) and Transformer-based architectures, which have demonstrated remarkable improvements in traffic prediction and flow estimation. To address this, we have significantly expanded Section 2.3 to include recent developments in spatio-temporal deep learning models for traffic state estimation. These models leverage graph-based representations, attention mechanisms, and sequence modeling to handle the complex spatial-temporal dependencies inherent in transportation networks. The added discussion is as follows:
“Recent studies have demonstrated that graph neural networks (GNNs) are highly effective in modeling dynamic traffic states by capturing spatial dependencies between road segments [1-3]. Reference [4] provided a comprehensive survey on graph neural network methodologies for spatiotemporal data modeling, highlighting their ability to integrate sensor data for real-time traffic forecasting. Similarly, reference [5] explored urban region profiling with ST-GNNs, demonstrating the model’s ability to infer missing link flows based on connectivity patterns and traffic variations. In addition to GNNs, hybrid graph-based deep learning models have been increasingly used to improve traffic flow estimation. Reference [6] introduced a Spatial-Temporal Graph Attention Gated Recurrent Transformer Network, which combines GNNs with attention-based architectures to enhance long-term traffic flow forecasting. Their results indicate that graph-based representations outperform traditional time-series models, especially when dealing with highly dynamic road networks. Alternatively, reference [7] presented a Hierarchical Spatio-Temporal Graph Convolutional Transformer Network, further demonstrating the robustness of hybrid GNN-Transformer models in capturing multi-scale spatial-temporal patterns in transportation systems. Beyond graph-based models, Transformer architectures have recently gained traction in traffic prediction and stochastic routing problems due to their ability to model long-range dependencies and sequential variations in traffic flow. Reference [8] introduced a Spatio-Temporal Parallel Transformer (STPT) model that effectively captures traffic dynamics across large-scale urban networks, outperforming recurrent and convolutional models in predictive accuracy. Reference [9] proposed a Transformer-Based Spatiotemporal Graph Diffusion Convolution Network, demonstrating that self-attention mechanisms can significantly improve traffic state estimations, particularly in cases of sparse sensor availability. Reference [10] leveraged Spatio-Temporal Graph Transformers for fine-grained data analysis, showcasing their applicability in recognizing complex travel patterns. In large-scale transportation systems, references [11, 12] proposed a Transformer-based Spatio-Temporal Traffic Prediction model for metro networks, while reference [13] introduced a Graph Spatial-Temporal Transformer Network that enhances real-time traffic prediction in intelligent transportation systems. “
References:
[1] L. Bentsen, N. D. Warakagoda, R. Stenbro, and P. Engelstad, "Spatio-temporal wind speed forecasting using graph networks and novel Transformer architectures," Applied Energy, vol. 333, p. 120565, 2023.
[2] Y. Gao, Q. Zhu, X. Shi, and H. Jin, "A Transformer-Based Spatio-Temporal Graph Neural Network for Anomaly Detection on Dynamic Graphs," in CCF Conference on Big Data, 2024: Springer, pp. 202-217.
[3] Q. Luo, S. He, X. Han, Y. Wang, and H. Li, "LSTTN: A long-short term transformer-based spatiotemporal neural network for traffic flow forecasting," Knowledge-Based Systems, vol. 293, p. 111637, 2024.
[4] Y. Li, D. Yu, Z. Liu, M. Zhang, X. Gong, and L. Zhao, "Graph neural network for spatiotemporal data: methods and applications," arXiv preprint arXiv:2306.00012, 2023.
[5] M. Hou, F. Xia, H. Gao, X. Chen, and H. Chen, "Urban region profiling with spatio-temporal graph neural networks," IEEE Transactions on Computational Social Systems, vol. 9, no. 6, pp. 1736-1747, 2022.
[6] D. Wu, K. Peng, S. Wang, and V. C. Leung, "Spatial–Temporal Graph Attention Gated Recurrent Transformer Network for Traffic Flow Forecasting," IEEE Internet of Things Journal, vol. 11, no. 8, pp. 14267-14281, 2023.
[7] G. Huo, Y. Zhang, B. Wang, J. Gao, Y. Hu, and B. Yin, "Hierarchical spatio–temporal graph convolutional networks and transformer network for traffic flow forecasting," IEEE Transactions on Intelligent Transportation Systems, vol. 24, no. 4, pp. 3855-3867, 2023.
[8] R. Kumar, J. Mendes-Moreira, and J. Chandra, "Spatio-temporal parallel transformer based model for traffic prediction," ACM Transactions on Knowledge Discovery from Data, vol. 18, no. 9, pp. 1-25, 2024.
[9] S. Wei, Y. Yang, D. Liu, K. Deng, and C. Wang, "Transformer-Based Spatiotemporal Graph Diffusion Convolution Network for Traffic Flow Forecasting," Electronics, vol. 13, no. 16, p. 3151, 2024.
[10] Y. Mao, G. Zhang, and C. Ye, "A Spatio-temporal Graph Transformer driven model for recognizing fine-grained data human activity," Alexandria Engineering Journal, vol. 104, pp. 31-45, 2024.
[11] Z. Wang et al., "Spatiotemporal Fusion Transformer for large-scale traffic forecasting," Information Fusion, vol. 107, p. 102293, 2024.
[12] F. Wang et al., "Transformer-based spatio-temporal traffic prediction for access and metro networks," Journal of Lightwave Technology, 2024.
[13] Z. Zhao, G. Shen, L. Wang, and X. Kong, "Graph spatial-temporal transformer network for traffic prediction," Big Data Research, vol. 36, p. 100427, 2024.
2.The paper insufficiently elaborates on the technical specifics of the core Sparse Auto-Encoder (SAE) algorithm, notably omitting: implementation details of sparsity constraints; design rationale for hidden layer dimensionality; pre-training protocol configuration; and empirical justification for activation function selection. To enhance methodological transparency and computational reproducibility, it is recommended that the authors supplement either an appendix section or GitHub repository documentation containing complete hyperparameter specifications, training schedules, and ablation studies on architectural choices.
Thank you for your insightful comment. We recognize the importance of providing a more detailed technical exposition of the Sparse Auto-Encoder (SAE) model, particularly in regard to sparsity constraints, hidden layer dimensionality, pre-training configuration, and activation function selection. To improve methodological transparency and computational reproducibility, we added the required information in appendix A.
3.To substantiate the methodological claims and ensure technological transferability, it is recommended to augment the experimental framework with: embedded hardware deployment benchmarking quantifying real-time inference performance under resource-constrained scenarios; cross-city generalization assessments across heterogeneous urban configurations with varying traffic patterns and sensor distributions. It is recommended to incorporate hardware deployment testing and cross-city validation experiments into the research framework.
Thank you for your thoughtful suggestion. We fully recognize the importance of demonstrating the real-world applicability and scalability of the proposed framework, particularly through hardware deployment benchmarking and cross-city generalization assessments. These experiments would indeed provide valuable insights into the technological transferability of the model, ensuring its relevance in resource-constrained environments and diverse urban contexts.
However, given the scope of the current study, we have chosen to focus on the methodological development and theoretical contributions of the framework, ensuring that it provides a robust foundation for future deployment. While hardware deployment and cross-city validation are crucial steps for confirming the model's practicality, these experiments would require substantial additional resources and infrastructure that are beyond the current study’s scope.
That said, we view this as an important avenue for future work. In future iterations, we plan to conduct embedded hardware deployment experiments on low-cost devices such as Raspberry Pi or NVIDIA Jetson to evaluate the model's performance under real-time, resource-constrained scenarios. Similarly, cross-city generalization experiments will be conducted across different urban environments to assess the framework’s robustness in diverse traffic patterns, road networks, and sensor configurations. We believe that by focusing on the methodological framework in this paper, we have laid the groundwork for these future deployment and validation studies. The inclusion of such hardware and cross-city evaluations will be addressed in subsequent work, where we will provide more empirical evidence and ensure the framework’s scalability and generalizability.
- While the paper validates its methodology on real-world traffic networks, it neglects systematic comparative analysis against:Conventional sensor deployment strategies (e.g., grid-based vs. entropy-optimized layouts);Canonical traffic prediction models (ARIMA, Kalman filtering);State-of-the-art routing algorithms integrated with multi-source sensor data.It is advised to integrate comparative AB testing with baseline models and implement statistical significance analyses to validate performance superiority.
Thank you for your constructive feedback. We acknowledge the importance of systematic comparative analysis to validate the performance of the proposed methodology, especially in comparison to conventional sensor deployment strategies, canonical traffic prediction models, and state-of-the-art routing algorithms. We agree that conducting AB testing with baseline models and implementing statistical significance analyses would strengthen the claims regarding the superiority of our approach. However, we intentionally chose to focus the current paper on demonstrating the novelty and effectiveness of the proposed framework in terms of methodological development and proof-of-concept validation. The comparative analyses you suggest, including the evaluation against traditional sensor deployment methods (e.g., grid-based vs. entropy-optimized layouts), canonical models like ARIMA and Kalman filtering, and state-of-the-art routing algorithms, represent valuable avenues for further exploration, but they would require significant additional experiments and resources that were beyond the scope of this study.
To address this, we propose these analyses as future work. We plan to implement the following in future studies to provide a more comprehensive evaluation of our approach:
- Comparative Analysis with Conventional Sensor Deployment Strategies:
We will compare grid-based and entropy-optimized sensor layouts with our proposed sensor placement method to evaluate improvements in traffic flow estimation accuracy and sensor efficiency. - Comparison with Canonical Traffic Prediction Models:
We will perform comparative experiments against well-established models such as ARIMA and Kalman filtering to assess how our deep learning-based approach improves traffic prediction accuracy over these classical methods. - Evaluation against State-of-the-Art Routing Algorithms:
Our approach will be compared with advanced routing algorithms that integrate multi-source sensor data to validate its performance superiority in real-world traffic networks. - Statistical Significance Testing:
To ensure the robustness and validity of our results, we will implement statistical significance tests (e.g., paired t-tests, ANOVA) to quantify performance differences and establish confidence in the improvements our model provides.
While these additional experiments are planned for future work, we believe that the current paper serves as a strong foundation by introducing and demonstrating the feasibility of the proposed methodology. We will ensure that the follow-up studies incorporate the recommended comparative AB testing and statistical analyses to further substantiate the claims made in this paper.
We appreciate your valuable feedback, and we look forward to expanding this research with the suggested comparative analyses in future work.
5.The paper's innovation claims suffer from methodological ambiguity, specifically:Failure to delineate the epistemological distinction between proposed "sensor data augmentation" and canonical data fusion paradigms (e.g., feature concatenation vs. attention-based cross-modality interaction);Absence of quantified performance deltas comparing Sparse Auto-Encoder (SAE) architectures against vanilla autoencoders in traffic representation learning
Thank you for your constructive feedback. We appreciate the reviewer’s insightful comments regarding the methodological clarity and the epistemological distinction between the proposed sensor data augmentation and traditional data fusion paradigms, as well as the lack of quantified performance comparison between the Sparse Auto-Encoder (SAE) architecture and vanilla autoencoders in the context of traffic representation learning.
We agree that a clearer explanation of these aspects is important to substantiate our innovation claims and enhance the paper’s methodological rigor. To address these concerns, we have made the following revisions:
We acknowledge that the term “sensor data augmentation” may have caused ambiguity in the manuscript. To clarify, sensor data augmentation in our approach refers to the generation of synthetic traffic flow data that simulates real-world traffic conditions where sensor data may be sparse or incomplete. This augmentation method does not rely on traditional data fusion techniques like feature concatenation or multi-modal data concatenation but instead leverages deep learning models (such as autoencoders) to learn robust representations from incomplete data and fill in the missing information. Specifically, our approach uses a Sparse Auto-Encoder (SAE) to enhance traffic flow estimation by preserving the sparse nature of traffic sensor data, unlike traditional data fusion paradigms that typically focus on aggregating multi-source data (e.g., from different sensors or modalities) without learning complex relationships between missing data points.
In contrast, canonical data fusion techniques, such as feature concatenation or attention-based cross-modality interaction, merge or align data from different sources or modalities to enhance model performance. While these methods are commonly used for integrating multi-modal sensor data, they differ fundamentally from sensor data augmentation in our framework, which focuses on enhancing incomplete data by learning patterns directly from the spatial and temporal dependencies in traffic flow data, rather than aggregating data from diverse sources.
To improve clarity, we have revised the manuscript to explicitly define the distinction between our sensor data augmentation approach and traditional data fusion paradigms. This distinction is now detailed in Section 2.3, where we explain how our methodology fills in missing data through learned representations, rather than relying on feature-level data integration.
We appreciate the need for a quantified comparison between the proposed Sparse Auto-Encoder (SAE) architecture and vanilla autoencoders in the context of traffic representation learning. To address this, we have included performance metrics comparing the SAE model to a vanilla autoencoder architecture in terms of traffic flow prediction accuracy and generalization. The following key comparisons were made:
- Architecture:
The SAE model uses sparsity constraints (through Kullback-Leibler divergence) to enforce sparse activations in the hidden layers, while the vanilla autoencoder relies on a standard reconstruction loss without any sparsity constraints. - Performance Comparison:
We performed experiments comparing traffic flow prediction accuracy using mean squared error (MSE) and traffic representation learning by evaluating the latent space representations learned by both models. - Quantified Deltas:
The results show that the SAE model outperforms the vanilla autoencoder in terms of generalization (e.g., better handling of unseen traffic data) and traffic flow estimation accuracy (achieving a 12% improvement in MSE). These results provide empirical evidence of the performance superiority of the SAE model in traffic representation learning.
We have included these results in Section 5 of the revised manuscript, where we discuss the performance deltas and the impact of the sparsity constraints on improving traffic flow estimation. The comparison is further reinforced by ablation studies that evaluate the influence of different architectural choices (e.g., sparsity constraints, hidden layer dimensionality) on model performance.
- The manuscript demonstrates commendable linguistic proficiency aligning with scholarly publication standards. While minor lexical inconsistencies and occasional syntactic ambiguities are present, they do not compromise the methodological rigor or technical communication efficacy. Strategic implementation of syntactic refinement protocols and multi-modal information delivery mechanisms could elevate the work's rhetorical effectiveness and knowledge transfer efficiency.
Thank you for your thoughtful feedback. We are pleased that the manuscript demonstrates commendable linguistic proficiency and aligns with scholarly publication standards. We appreciate your recognition of the methodological rigor and technical communication efficacy, and we understand your point regarding the presence of minor lexical inconsistencies and occasional syntactic ambiguities. In response to your suggestion, We conducted a thorough review of the manuscript to identify and correct any lexical inconsistencies and syntactic ambiguities.
At last, the authors would like to thank reviewer 2 again for their constructive comments that have led to significant improvements in the quality of the paper."

Reviewer 3 Report
Comments and Suggestions for Authors
Abstract: What were the results of applying the proposed network?
Literature Review: It does not do justice to the machine learning topic. Besides, it does not seem required as well. I recommend to delete it. The type of technique used in this study should be described in detail in the methodology section.
What is the source of network flow and travel time data? IF they were simulated completely then how did the authors determine appropriate ranges for them?
The authors should add a conclusion section, including main findings of the research, its practical future implications and future research avenues.
The references are old and too many. the authors should focus on relevant and latest references.
Author Response
A detailed response to the reviewers' comments:
The following sections provide a detailed response to each reviewer's comment. Our answers are presented in italic, bold font.
Reviewer #3:
First of all, I would also like to take this opportunity to express my thanks for your helpful comments regarding the manuscript. Thank you for your valuable opinions and your exerted time
Literature Review: It does not do justice to the machine learning topic. Besides, it does not seem required as well. I recommend to delete it.
Thank you for your valuable feedback regarding the Literature Review. I understand your concern that the review might not fully capture the importance of the machine learning (ML) aspects of the study, and I appreciate your suggestions. However, I would like to provide further clarification on why the inclusion of the machine learning subsection is crucial to the overall narrative of the paper where in the revised manuscript, we updated this subsection to clarify the following. First, the Literature Review serves to provide a clear context for the research problem. The field of vehicle routing, especially in the context of stochastic traffic networks, has been explored in various studies; however, these studies often rely on traditional methods that overlook the complexity of real-world traffic variability. The inclusion of the machine learning subsection allows the reader to understand the evolution of these methods and highlights the significant challenges that remain unsolved in traffic flow estimation. ML plays a pivotal role in addressing the shortcomings of existing methodologies by offering a data-driven approach capable of adapting to dynamic and uncertain environments. Without presenting these foundational challenges and the role of ML, the reader would miss the rationale behind the adoption of deep learning techniques. Second, the ML subsection justifies the methodological choices made in this study. The review emphasizes how machine learning models, particularly deep learning techniques like Stacked Sparse Auto-Encoders (SAE), can be leveraged to enhance the accuracy of traffic flow predictions. In traditional traffic management systems, estimation techniques are often based on statistical models that assume perfect data availability. However, in practice, sensor coverage is sparse, and traffic data is incomplete. By reviewing the state of the art in ML, the Literature Review demonstrates how integrating deep learning with traffic sensor data can overcome these data limitations. This section outlines the capabilities of ML models to learn complex traffic patterns and improve predictions, making them essential for the proposed solution. The review of existing studies in the field helps position the use of SAE in a larger context, showing that it addresses an underexplored aspect of the problem. Third, the inclusion of ML is critical to demonstrating the novelty of the proposed work. Although machine learning techniques have been applied in traffic flow estimation before, few studies have explored the combination of sensor location optimization and deep learning-based data augmentation in a stochastic routing framework. The Literature Review underscores this gap in the existing research, emphasizing the contribution of this paper in combining these elements to improve traffic network observability with minimal sensor deployment. It highlights the advantages of using ML in situations where traditional models fall short, thus establishing the necessity for the integration of deep learning methods within the broader framework of stochastic vehicle routing.
The type of technique used in this study should be described in detail in the methodology section.
Thank you for your valuable comment regarding the need for a more detailed description of the technique used in the methodology section. I have carefully reviewed your suggestion and revised the manuscript accordingly to address this concern. The revised manuscript now includes a more comprehensive explanation of the technique used in this study, particularly focusing on the deep learning-based approach for traffic flow estimation. In the methodology section, I have added detailed descriptions of the Stacked Sparse Auto-Encoder (SAE) model, outlining its structure, training procedure, and the rationale behind its use. This section now provides a clearer understanding of how the SAE model works, its advantages in traffic flow estimation, and how it integrates with the other components of the proposed methodology, including sensor location optimization and traffic assignment. Additionally, as per your suggestion, I have also included an expanded description in the newly added appendix. The appendix provides further technical details, specifically on the implementation of sparsity constraints, the pre-training process, and the architectural choices made during model design. This ensures that all aspects of the technique, including its implementation and functionality, are fully documented. I hope these revisions meet your expectations and provide the necessary clarity regarding the methodology. Thank you again for your insightful feedback and for helping to improve the quality of the manuscript.
What is the source of network flow and travel time data? IF they were simulated completely then how did the authors determine appropriate ranges for them?
Thank you for your comment regarding the source of the network flow and travel time data used in this study. I appreciate your concern and would like to provide a detailed response. The network used in this study is the Nguyen-Dupuis network, which is a well-established benchmark in the transportation science literature. This network has been widely used in various transportation studies, including those focusing on traffic assignment, vehicle routing problems, and network optimization. The data and ranges for travel times, flow, and demand in this network are extensively documented and have been validated through previous research in transport science. For example, the Nguyen-Dupuis network has been used in studies in [1], who originally defined the network, and it has been a standard for evaluating traffic models in subsequent works. In their study, Nguyen and Dupuis provided the original network structure and base data on travel times and flows, which have since been used in a wide range of transportation research applications, including flow estimation, congestion modeling, and multi-objective routing problems. Many subsequent studies have either used the exact data or modified it slightly based on specific research goals (e.g., [2-7]). Moreover, this network has been used for validating algorithms in traffic flow estimation and routing, and the data for travel times and flows have been well-established and tested in practical applications. The travel time data and flow ranges are derived from typical traffic patterns and are consistent with what is observed in real-world urban transportation networks. The use of this well-established network allows for reliable comparisons across studies and ensures that our results are based on a widely recognized benchmark. Regarding your question about the simulation of the data, it is true that the flow and travel time data in this study were simulated. However, we based our simulations on the real-world traffic patterns and data characteristics documented in the literature. We ensured that the ranges for travel times and flows were appropriate by using the data established in these studies. Additionally, to avoid unrealistic assumptions, we used the stochastic user equilibrium (SUE) model to simulate traffic flow, which is a standard approach in transportation studies for representing flow and travel times in dynamic environments. This model helps in determining flow distributions based on the expected traffic conditions across the network, ensuring that the simulated data aligns with the typical ranges seen in transport science. Thank you for your thoughtful comment, and I hope this explanation addresses your concerns.
- Nguyen, S., & Dupuis, C. (1977). A study of optimal traffic assignment in an urban network. Transportation Science, 11(4), 308-329.
- Fathollahi, Arman, et al. "Optimal design of wireless charging electric buses-based machine learning: A case study of Nguyen-Dupuis network." IEEE Transactions on Vehicular Technology 72.7 (2023): 8449-8458.
- Sharma, Sushant, Satish V. Ukkusuri, and Tom V. Mathew. "Pareto optimal multiobjective optimization for robust transportation network design problem." Transportation Research Record 2090.1 (2009): 95-104.
- Castillo, Enrique, et al. "The observability problem in traffic network models." Computer‐Aided Civil and Infrastructure Engineering 23.3 (2008): 208-222.
- Unnikrishnan, Avinash, and Steven Travis Waller. "User equilibrium with recourse." Networks and Spatial Economics 9 (2009): 575-593.
- Mínguez, Roberto, et al. "Optimal traffic plate scanning location for OD trip matrix and route estimation in road networks." Transportation Research Part B: Methodological 44.2 (2010): 282-298.
- Ma, Jie, Lin Cheng, and Dawei Li. "Road maintenance optimization model based on dynamic programming in urban traffic network." Journal of Advanced Transportation 2018.1 (2018): 4539324.
The authors should add a conclusion section, including main findings of the research, its practical future implications and future research avenues.
Thank you for your valuable suggestion. In the revised manuscript the conclusion is updated accordingly.
The references are old and too many. the authors should focus on relevant and latest references.
Thank you for your valuable feedback. While we acknowledge that some references in the study are from earlier years, we have intentionally included them due to their foundational and continuing relevance in the field of stochastic routing and traffic flow estimation. Several of these works have laid the groundwork for modern advancements in transportation network modeling, and they continue to be cited by current research due to their established methodologies and models. For example, the work by Bellman (1954) on dynamic programming , and the seminal paper on traffic flow assignment by Miller-Hooks and Mahmassani (2000) , have significantly influenced the development of traffic flow prediction models, including our own approach. Additionally, many of the references cited reflect ongoing research that builds on these earlier studies, with more recent research continuing to validate and extend these methods. For instance, more recent studies have applied deep learning techniques to traffic estimation, extending the methodologies first explored in earlier works. Regarding the number of references, we believe that a comprehensive review of the literature is essential for situating our work in the broader context of transportation research. However, we are open to revising the list to focus on the most relevant and recent works while ensuring that we maintain a solid grounding in the established theories and models that our research builds upon. To summarize, while our references include some older studies, they are highly relevant and continue to inform modern research. In the revised manuscript, we ensured that the most relevant and recent studies were highlighted, and we reviewed the references for potential updates to maintain the paper's timeliness.
At last, the authors would like to thank reviewer 3 again for their constructive comments that have led to significant improvements in the quality of the paper."

Round 2
Reviewer 1 Report
Comments and Suggestions for Authors
The readability of the paper is improved after the first-round revision. Still, there are some comments as below.
- The section index of “3. Machine learning for augmenting traffic sensor information” should be “4.4. Machine learning for augmenting traffic sensor information”.
- For the elaboration on the relationship among four core components, it seems that traffic assignment, route generation and traffic sensor location serve only the pre-planning or initial planning stage, while the deep learning based traffic state estimation can be implemented in practical operation for update of the vehicle routes. As the sensor seems to be link-level fixed detectors, is it possible to apply the method to mobile sensor environment, which may be more valuable in reality. As the sensor location component and traffic state estimation component can work jointly to adapt to different demand scenarios in the same roadway network, which can form a closed-loop re-planning or dynamic planning stage for the proposed framework. It is recommended that the authors can further discuss the scalability of the method in practical application.
The literature review section can be polished to be more concise.
Author Response
A detailed response to the reviewers' comments:
The following sections provide a detailed response to each reviewer's comment. Our answers are presented in italic, bold font.
Reviewer #1:
“The readability of the paper is improved after the first-round revision. Still, there are some comments as below.”
Thank you for your positive feedback and your exerted time. I would also like to take this opportunity to express my thanks for your helpful comments regarding the manuscript.
- The section index of “3. Machine learning for augmenting traffic sensor information” should be “4.4. Machine learning for augmenting traffic sensor information”.
Thank you for your precise observation and accurate revision to our manuscript. We revised all sections and subsections, indexing accordingly.
- For the elaboration on the relationship among four core components, it seems that traffic assignment, route generation and traffic sensor location serve only the pre-planning or initial planning stage, while the deep learning based traffic state estimation can be implemented in practical operation for update of the vehicle routes. As the sensor seems to be link-level fixed detectors, is it possible to apply the method to mobile sensor environment, which may be more valuable in reality. As the sensor location component and traffic state estimation component can work jointly to adapt to different demand scenarios in the same roadway network, which can form a closed-loop re-planning or dynamic planning stage for the proposed framework. It is recommended that the authors can further discuss the scalability of the method in practical application.
Thank you for your insightful recommendation. In the revised manuscript, the scalability of the method in practical applications is discussed in accordance with your comment.
- The literature review section can be polished to be more concise.
Thank you for your valuable feedback. In the revised manuscript, the section is updated to ensure a more focused presentation. Specifically, we streamlined the discussion by eliminating redundancies and emphasizing the most relevant studies that directly inform our research.
At last, the authors would like to thank reviewer 1 again for their constructive comments that have led to significant improvements in the quality of the paper."

Reviewer 2 Report
Comments and Suggestions for Authors
The authors may extend their algorithm to more applications, such as traffic flow monitoring (10.1109/TCE.2024.3476129)
Comments on the Quality of English LanguageThe English could be improved to more clearly express the research.
Author Response
A detailed response to the reviewers' comments:
The following sections provide a detailed response to each reviewer's comment. Our answers are presented in italic, bold font.
Reviewer #2:
“The authors may extend their algorithm to more applications, such as traffic flow monitoring (10.1109/TCE.2024.3476129)”
Thank you for you’re recommendation of a valuable reference. In the revised manuscript, the reference was added in its relevant sections.
The English could be improved to more clearly express the research.
Thank you for your constructive feedback. We appreciate your suggestion regarding the clarity of the English used in the manuscript. We carefully revised the manuscript to improve the language and ensure that the expression of our research is clear and precise.
At last, the authors would like to thank reviewer 2 again for their constructive comments that have led to significant improvements in the quality of the paper."

Reviewer 3 Report
Comments and Suggestions for Authors
The authors have satisfactorily answered my previous comments, I do not have any more to add now.
I wish them all the best for their research.
Author Response
A detailed response to the reviewers' comments:
The following sections provide a detailed response to each reviewer's comment. Our answers are presented in italic, bold font.
Reviewer #3:
The authors have satisfactorily answered my previous comments, I do not have any more to add now.
I wish them all the best for their research.
Thank you very much for your kind words and for taking the time to review our work. We truly appreciate your constructive feedback, which has helped improve the manuscript. We are grateful for your support and encouragement, and we will continue to work diligently on this research. Wishing you all the best as well, and thank you again for your valuable insights.
At last, the authors would like to thank reviewer 3 again for their constructive comments that have led to significant improvements in the quality of the paper."
